

# Cavity-induced bifurcation in classical rate theory

**Kalle Sulo Ukko Kansanen⋆ and Tero Tapio Heikkilä**

Department of Physics and Nanoscience Center, University of Jyväskylä, Finland

⋆ kalle.kansanen@gmail.com

## Abstract

We show how coupling an ensemble of bistable systems to a common cavity field affects the collective stochastic behavior of this ensemble. In particular, the cavity provides an effective interaction between the systems, and parametrically modifies the transition rates between the metastable states. We predict that the cavity induces a phase transition at a critical temperature which depends linearly on the number of systems. It shows up as a spontaneous symmetry breaking where the stationary states of the bistable system bifurcate. We observe that the transition rates slow down independently of the phase transition, but the rate modification vanishes for alternating signs of the system–cavity couplings, corresponding to a disordered ensemble of dipoles. Our results are of particular relevance in polaritonic chemistry where the presence of a cavity has been suggested to affect chemical reactions.

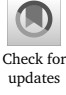

# 1  Introduction

Classical rate theory provides a stochastic framework for investigations in natural and social sciences [1–6]. It has been used to describe a wide variety of phenomena in different fields: the spreading of diseases in epidemiology [7,8], the growth and decline of companies in economics [9] and of populations in ecology [10], and disintegration of radioactive nuclei in atomic physics, to name a few. In chemistry, rate equations provide a quantitative way of understanding chemical reactions [11].

There have been several recent indications of chemical reactions being altered by the formation of vibrational polaritons, hybrid excitations formed by the coupling of molecular vibrations to vacuum electromagnetic field of an optical cavity [12–14]. The vacuum field has been reported to work both as a catalyst [15,16] and an inhibitor [17–20] in different reactions. However, the current theoretical understanding seems to contradict these experimental results. The conventional approach that concentrates on the cavity-induced modification of the individual reaction rates within the transition state theory finds that all the polaritonic effects should disappear when the number of molecules forming the polariton increases [21–25]. Furthermore, very recently, there have been reports of failed replication experiments [26,27], adding to the confusion.

Regardless of the current experimental status, vacuum-modified chemistry poses a fundamental challenge on our understanding of light-matter interactions. Its difficulty arises from the notion of collectivity. The cavity fields interact with a large number of molecules which forms an interacting many-body system. The collectivity of the light-matter system leads to novel optical properties [28], including the realization of hybrid Bose-Einstein condensates [29,30] and sub-/superradiant quantum emitters [31], as well as novel interactions whose role in materials design is actively studied [32–36].

In this article, we predict a phase transition induced by the presence of the cavity. This transition is caused by the indirect interactions within the matter system, e.g. molecules, mediated by the vacuum field. We show that such an interaction can be taken into account in the classical rate theory as transition rates that depend parametrically on the state of the molecules. We find an order parameter for the light-matter system and show that, below a certain critical temperature which depends linearly on the number of molecules, it bifurcates so that it can have multiple solutions in the stationary state. The phase transition aligns the molecular states depending on their coupling to the cavity. If the molecules couple equally, they will spontaneously prefer being in the same state. When this is not the case, there is a phase

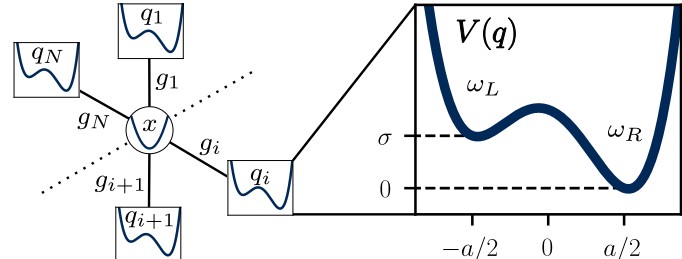

Figure 1: Interactions between a harmonic cavity mode $x$ and $N$ identical modes $q_i$, described by a bistable potential $V(q)$. In the harmonic approximation, $\partial_q^2 V(a/2) = \omega_R^2$ and $\partial_q^2 V(-a/2) = \omega_L^2$ define the relevant frequencies.

separation based on the light-matter coupling constants. Even before the phase transition, the cavity-mediated interaction may slow down the effective transition rates, suggesting that the collectivity of the many molecule system must be considered on the level of the rate equations, not only on the level of the rates themselves.

## 2 Rate theory

Let us consider a one-dimensional and bistable system described by a potential $V(q)$ as in Fig. 1. Here, $q$ is a position quadrature which can describe, e.g., vibrational modes of a molecule in its electronic ground state. Classical rate theory then relates the probability $p$ to find the system in one of the wells to the transition rates $\Gamma_{\mu\nu}$ between the wells. We adopt a terminology of left and right wells corresponding to the minima of $V(q)$ at $q < 0$ and $q > 0$, respectively. The state of $N$ bistable systems can be described with the vector $(s_1, s_2, \ldots, s_N)$ where $s_j = \text{sign}(q_j)$. Thus, if $p_j$ is the probability to find the $j$th system in the left well ($s_j = -1$), its rate equation[1] reads as

$$\frac{d}{dt} p_j = -\Gamma_{LR}^j p_j + \Gamma_{RL}^j \left[ 1 - p_j \right]. \tag{1}$$

The macroscopic behaviour is obtained from Eq. (1) by noting that the expectation value of the number of systems in the left well is given by $N_L = \sum_j p_j$. If the transition rates are independent of the system $j$, we find

$$\frac{d}{dt} \frac{N_L}{N} = -\Gamma_{LR} \frac{N_L}{N} + \Gamma_{RL} \left[ 1 - \frac{N_L}{N} \right]. \tag{2}$$

In the following, we show that this approach has to be extended when the molecules have varying couplings to the cavity.

Often, the transition rates are assumed to be independent of $p_j$ and time, i.e., constants. Physically, this is motivated by separation of time scales. We assume the transitions to happen so rarely that the molecules manage to relax and their environment lose any information before the next transition [11]. We consider the thermal activation regime, the ratio between the potential energy barrier and the temperature $k_B T$ controls the rate of transitions and should then be sufficiently large. This is in contrast with the quantum tunneling regime where the energy barrier height is compared to the resonant frequency inside the potential well [25]. In such a case, to comply with thermodynamics, the rates must obey the detailed balance $\Gamma_{LR}/\Gamma_{RL} = e^{\beta\sigma}$ where $\beta = 1/k_B T$ is the inverse temperature and $\sigma = V(-a/2) - V(a/2)$

---

[1]We use the term "rate equation" here. In other contexts, at least the names "Kolmogorov equation" and "master equation" have been used to describe the same approach.

corresponds to the potential bias between the different states (Fig. 1). The rate equation then describes thermalization to the Boltzmann distribution with $p_j = N_L/N = 1/(1 + e^{\beta\sigma})$.

The coupling of the molecules to the vacuum field leads to a state-dependent modification of the rates in Eq. (1). This is caused by the effective interaction between the different molecules mediated by the cavity. Before deriving the modification explicitly, we write this statement formally as $\Gamma^j_{\mu\nu} = \Gamma^0_{\mu\nu} r^j_{\mu\nu}(N_L)$ where $r^j_{\mu\nu}(N_L)$ represents the rate modification caused by the vacuum field which may be expected to be a function of $N_L$ and not of the individual $p_j$'s. The subsequent solution of the rate modifications $r^j_{\mu\nu}$ is the main result of this work from which the cavity-induced collective effects follow.

## 3 State-dependent rate modification

Consider $N$ systems with coordinates $q_j$ coupled to an external collective mode as in Fig. 1. In what follows, we call those $N$ systems "molecules" and the collective mode is a "cavity", in analogy to the systems studied in polaritonic chemistry. However, this approach also works, for example, in the case of a large number of superconducting flux qubits coupled to a single microwave cavity, or bistable atoms in a cold-atom arrangement, coupled to a common cavity mode. These potential realizations are discussed in Appendix G.

We derive the state-dependent transition rates from the detailed balance. That is, if the stationary state of the total system thermalizes in the presence of the cavity, the rates for the molecule $j$ obey

$$\frac{\Gamma^j_{LR}}{\Gamma^j_{RL}} = \exp\left\{\beta\left[V_{\text{tot}}(\ldots,s_{j-1},-1,s_{j+1},\ldots) - V_{\text{tot}}(\ldots,s_{j-1},+1,s_{j+1},\ldots)\right]\right\}, \qquad (3)$$

when $V_{\text{tot}}$ is the potential energy for a given state of the molecule-cavity system. The quantity in square brackets represents the energy cost of moving one molecule from right to left well.

Motivated by quantum electrodynamics (QED) in the dipole gauge [23,37–41], we introduce a single cavity mode of frequency $\omega_c$ and its electric field quadrature $x$ and choose the classical potential energy of the system[2] to be similar to that producing polariton excitations,

$$V_{\text{tot}}(x,q_1,\ldots,q_N) = \sum_{j=1}^N V(q_j) + \frac{1}{2}\omega_c^2 x^2 + \sum_{j=1}^N d_j(q_j)x, \qquad (4)$$

where $d_j(q_j)$ represents the molecular dipole moment component in the direction of the cavity mode's polarization vector. In the Born–Oppenheimer approximation, $d_j(q_j)$ is proportional to the expectation value of the dipole operator in the electronic ground state with a given value of $q_j$. We note that the energy does not include the so-called dipole self-energy terms, proportional to $[d_j(q_j)]^2$ in the long-wavelength limit, similar to other recent works in strong coupling regime [20–22, 41]. Moreover, the model containing only a single cavity mode is chosen for simplicity. Generalizing the results to the case of several cavity modes is relatively straightforward and does not lead to qualitative changes in the results. Quantitatively, this would amount to calculating the Casimir-Polder-van der Waals [42] interaction between the dipoles in a cavity. Note that such an interaction is present also without the presence of a cavity: we assume that without the cavity the interaction is to be too small to lead to the collective effects discussed here.

We focus on a linear dipole moment $d_j(q_j) = \sqrt{\omega_c\omega_m}g_jq_j$ to gain insight on the effect of the cavity on the transition rates. Here, $\omega_m^2 = (\omega_L^2 + \omega_R^2)/2$ represents the mean frequency of

---

[2]The quadratures are chosen so that the corresponding kinetic energy is $(p_x^2 + \sum_i p_i^2)/2$, i.e., all the information about polaritonics resides in the potential alone.

the potential $V(q)$, and the square root term normalizes the light-matter coupling constant $g_j$. These choices allow for mapping this model potential to the Dicke Hamiltonian in which case $g_j$ follows directly from the QED derivation [43]. However, our approach works for an arbitrary dipole function $d_j(q_j)$ with the linear approximation $d_j(q_j) \approx d_j(\pm a/2) + d_j'(\pm a/2)[q_j \mp a/2]$ near the minima of $V(q_j)$. The important parameters are then the transition dipole moments $d_j'(\pm a/2)$, responsible for coupling molecules in left and right wells to light, and the difference in the dipole moments $d_j(a/2) - d_j(-a/2)$ between the metastable states. For a linear dipole moment, these three parameters are determined by $g_j$, simplifying the subsequent analysis.

The light-matter coupling $g_j$ depends on both the position of the molecule within the cavity and the alignment of its dipole moment with respect to the cavity polarization vector [43–45]. Importantly, $g_j$ can be either positive or negative which has physical consequences for the many molecule system. The strength of the collective coupling is indicated by the Rabi splitting frequency $\Omega \propto \sqrt{\sum_j g_j^2}$, defined as the difference of the eigenfrequencies of the total potential on resonance $\omega_L = \omega_R = \omega_c$ [46,47]. We assume that $g_j \ll \omega_c, \omega_m$ while $\Omega$ is a small fraction of $\omega_c$ and $\omega_m$ due to $N \gg 1$.

The effective interaction mediated by the cavity can be understood by a force analysis. For a system in a stationary state, displacing the mode $q_j$ by $\delta q_j$ causes a force proportional to $-g_j \delta q_j$ to the cavity mode $x$. This leads to a displacement $\delta x$ proportional to the force to the cavity mode. The force applied on another mode $q_k$ is then proportional to $-g_k \delta x$ which reads as $g_k g_j \delta q_j$ in terms of the original displacement on mode $q_j$. Whenever the coupling constants of two different molecules share the same sign, the force between them is attractive and for opposite signs repulsive. The cavity-induced interaction resembles the phonon-mediated interaction between electrons in superconductivity [48].

We solve the stationary states of $V_{\text{tot}}(x, q_1, \ldots, q_N)$ within the harmonic approximation which holds when $g_j \ll \omega_c, \omega_m$ and, thus, the cavity-induced change to the local minimum points of $V(q_j)$ is small. Technically, we first solve the values of $q_j$'s and $x$ from the conditions $\partial_x V_{\text{tot}} = \partial_{q_j} V_{\text{tot}} = 0$ by treating the signs of $q_j$ as variables, representing whether the molecule $j$ is in the left or the right well. We find that all the stationary points may be expressed by using the collective order parameter

$$\Delta = \frac{1}{N} \sum_{i=1}^{N} \frac{g_i}{\sqrt{\langle g^2 \rangle}} s_i, \tag{5}$$

where $\langle g^2 \rangle = \sum_i g_i^2 / N$ represents the average over the molecules and $s_i = \text{sign}(q_i)$. If there is no variation in the coupling constants $g_i$, $\Delta$ becomes the normalized difference between the number of molecules in the right and the left well, $\Delta \to \delta N / N = (N_R - N_L)/N$. Finally, we calculate the value of $V_{\text{tot}}(x, q_1, \ldots, q_N)$ using Eq. (4) at these stationary points. The details of this algebraic calculation with an arbitrary dipole moment function $d_j(q_j)$ are relegated to Appendix A. We note here that the order parameter $\Delta$ has the same form if one replaces $g_i$ with state-dependent light-matter couplings $g_i(s_i)$ arising from transition dipole moments $d'(\pm a/2)$, but the effects reported below disappear if the dipole moment is the same on both states, i.e., $d_i(a/2) - d_i(-a/2) = 0$.

We find that the stationary states of the full $N$ molecule system have the potential energy

$$\frac{V_{\text{tot}}(s_1, \ldots, s_N)}{N} = -E_b P \Delta^2 - \frac{\sigma}{2} \frac{\delta N}{N}. \tag{6}$$

The first term is caused by the effective interaction via the cavity mode while the second term describes the potential bias of $V(q)$. The magnitude of the interaction is determined by the

product of an effective potential barrier $E_b = \frac{1}{2}\omega_m^2(a/2)^2$ [3] and

$$P = \frac{N\langle g^2\rangle}{\omega_c\omega_m - \sum_i(\frac{\omega_m}{\omega(s_i)}g_i)^2} \approx \frac{N\langle g^2\rangle}{\omega_c\omega_m - N\langle g^2\rangle}. \tag{7}$$

Here, $\omega(s_i)$ equals $\omega_L$ for $s_i = -1$ and $\omega_R$ for $s_i = +1$. The coefficient $P$ can be related to the Rabi splitting $\Omega$ using $N\langle g^2\rangle \propto \Omega^2$. We assume that $\Omega$ is a small fraction of $\sqrt{\omega_c\omega_m}$ and $P \ll 1$. Consequently, $P$ may be treated as a state-independent constant since its variation is proportional to $P^2$ as shown in Appendix B.

Using the detailed balance (3) and Eq. (6), we find in the lowest order of $P$ that

$$\frac{\Gamma_{LR}^j}{\Gamma_{RL}^j} = \frac{\Gamma_{LR}^0}{\Gamma_{RL}^0}\frac{r_{LR}^j}{r_{RL}^j} = \exp\left(\beta\sigma + 2\alpha\frac{g_j}{\sqrt{\langle g^2\rangle}}\Delta\right), \tag{8}$$

where $\alpha = 2P\beta E_b$ plays the role of a control parameter. It should be noted that, even though $P \ll 1$, the effective potential barrier $E_b$ can be much larger than the temperature so that $\alpha$ can be of the order of unity.

Equation (8) expresses the relative change of the rates due to the change of the number of molecules in the left and right wells. To find the individual rates, we use the fact that reversing all coordinates also interchanges the rates. That is, we impose that $r_{RL}^j$ must be equal to $r_{LR}^j$ if we at the same time transform $\sigma \to -\sigma$, $\omega_{L/R} \to \omega_{R/L}$, and $q_i \to -q_i$ for every $i$. The control parameter $\alpha$ retains its value in this parity transformation. By using this equality in Eq. (8) and expanding the exponent of $r_{LR}^j$ in the powers of $\Delta$, we find that

$$r_{LR}^j = C\exp\left(\alpha\frac{g_j}{\sqrt{\langle g^2\rangle}}\Delta + f\right), \tag{9}$$

where $f$ is a function of $\Delta$ that stays invariant in the parity transformation. A few examples of such functions are $\Delta^2$ and $\sigma\Delta$. $C$ is a constant that describes a state-independent cavity-induced modification that may be acquired in, e.g., the transition state theory. Here, we assume that such modifications are small and $C = 1$. The function $f$, although not forbidden by the symmetry, we neglect as there is no clear physical origin for such terms. That is, we set $f = 0$. In this case, $r_{RL}^j = 1/r_{LR}^j$.

Next, we use Eq. (8) to evaluate the stationary state of the macroscopic system. We note that our approach is consistent with thermodynamics by construction, and hence the stationary properties could also be found by treating the light-matter system as a canonical ensemble and solving its thermal distribution. The separate rate modifications (9) allow for a continuous-time Markov chain simulation of Eq. (1) from which dynamics can also be investigated and compared to an analytical approach.[4] We further elaborate the numerical method in Appendix E.

# 4 Cavity-induced phase transition and collective dynamics

## 4.1 Equal light-matter coupling strengths

We illustrate some qualitative consequences of the parametric rate modifications by taking a simplifying limit of identical couplings as in [46]; we set $g_j = g_0$ for all $j$. In this case, the

---

[3]Note that $E_b$ is typically larger than the true potential barrier energy that determines the transition rates $\Gamma_{\mu\nu}^0$ in the absence of the cavity.

[4]The numerical simulation code and all plotting scripts are based on Python (especially, numpy-, scipy, and matplotlib-packages) and can be found at https://gitlab.jyu.fi/jyucmt/prt2022.

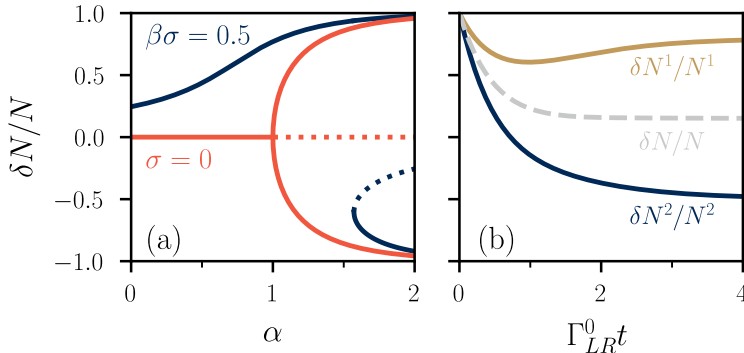

Figure 2: (a) Steady state solutions of Eq. (2) exhibiting bifurcation for equal couplings. The dotted line represents an unstable solution whereas the solid lines are stable; the average tends to one of the stable solutions after a long time. (b) Phase separation of a polaritonic system that is divided into two equally large partitions ($N^1 = N^2 = N/2$) with couplings related by $g_1 = -2g_2$. The solid lines represent the dynamics of the partitions alone whereas the dashed line describes the dynamics of the full system. Here, the potential $V(q)$ is symmetric with $\Gamma^0_{LR} = \Gamma^0_{RL}$ and $\sigma = 0$. The system has undergone the phase transition as $\alpha = 1.3$.

transition rates become independent of the molecule in question. The set of rate equations can be mapped to a master equation describing the probability to find exactly $N_L$ molecules in the left well as shown in Appendix D. Interestingly, a similar state-dependent master equation has been used to model the formation of time crystals in magneto-optical traps [49]. We confirm numerically that such a master equation and the macroscopic rate equation (2) produce the same results as long as $N \gg 1$.

We find the stationary state of the full $N$ molecule system by inserting Eq. (8) into the macroscopic rate equation (2) together with $\frac{d}{dt}N_L = 0$. It gives

$$\frac{\delta N}{N} = \tanh\left(\beta\frac{\sigma}{2} + \alpha\frac{\delta N}{N}\right). \tag{10}$$

The solutions for $\delta N$ determine the long-time behaviour of the macroscopic system.

First, let us consider $\sigma = 0$. If there is no coupling to light, the only stationary state is $N_L = N_R = N/2$ due to the left/right symmetry. For small values of $\alpha = 2P\beta E_b$ there is no change to this stationary state. However, as soon as $\alpha > 1$, we find two new stationary states while the state $N_L = N_R$ becomes unstable. The stationary states are represented in Fig. 2(a) as a function of the control parameter $\alpha$. As this configuration is possible only if the coupling is large enough, we call it the cavity-induced bifurcation of the stationary state.

The cavity-induced bifurcation remains in the presence of a finite bias $\sigma$. However, the critical value of $\alpha$ increases as $\sigma$ increases. For a very large bias $\beta\sigma \gg \alpha$, all the systems eventually end up in the right well which is lower in energy. This is expected, as biasing the potential pushes the molecules towards the right well whereas the attractive interaction tries to keep the molecules in the same well. Before the bifurcation, one can also see the effect of the cavity-mediated attraction in that more molecules can be found in the right well than without the cavity. This seems as if the cavity increased the potential bias.

The mere state-dependent light-matter coupling — caused by a nonlinear $d_i(q_i)$ — may bias the system. These effects are described at length in Appendix F. Similar to a potential bias, the molecules prefer the state with the larger coupling strength even with a single stationary state, and the larger the coupling difference, the larger the critical value of $\alpha$.

## 4.2 Effect of coupling strength distribution

If there is only a partial control of the coupling constants as is typical in polaritonic chemistry, one must include the variation of the couplings. We argue that this can but does not necessarily remove the cavity induced effects. Especially, the role of the average $\langle g \rangle$ is important. This average is independent of the optically observed Rabi splitting which is proportional to $\sqrt{N \langle g^2 \rangle}$ [12–14, 50, 51] and hence does not depend on the sign of the individual $g_j$'s.

The order parameter $\Delta$ determines the individual rates in Eq. (8). If we consider a partition of the $N$ molecules such that all molecules within the partition share the same coupling constant $g_j$, they also share the same transition rates $\Gamma^j_{\mu\nu}$. Macroscopic rate equations similar to Eq. (2) can be derived for each partition but deriving a closed-form differential equation for $\Delta$ is impossible. Consequently, the full macroscopic rate equation cannot be evaluated. However, the stationary states of the partitions can be solved in terms of $\Delta$. As shown in Appendix C, this leads to a self-consistency equation

$$\Delta = \left\langle \frac{g}{\sqrt{\langle g^2 \rangle}} \tanh\left( \beta \frac{\sigma}{2} + \alpha \frac{g}{\sqrt{\langle g^2 \rangle}} \Delta \right) \right\rangle, \tag{11}$$

whose solution can be used to find

$$\frac{\delta N}{N} = \left\langle \tanh\left( \beta \frac{\sigma}{2} + \alpha \frac{g}{\sqrt{\langle g^2 \rangle}} \Delta \right) \right\rangle, \tag{12}$$

given a distribution of coupling constants $g$.

We specifically discuss the case $\sigma = 0$. By expanding Eq. (11) to the third order in $\Delta$, one finds the solutions $\Delta \propto \pm\sqrt{\alpha - 1}$ and $\Delta = 0$. Thus, $\alpha = 1$ is the critical value for the bifurcation of the order parameter $\Delta$. This agrees with Fig. 2 in which $\Delta = \delta N / N$. The critical temperature of the phase transition is approximately given by

$$T_c(\sigma = 0) \approx \frac{2E_b}{k_B} \frac{N \langle g^2 \rangle}{\omega_c \omega_m}. \tag{13}$$

At the same time, Eq. (12) may be linearized giving $\delta N \propto \langle g \rangle \Delta$. Whenever $\langle g \rangle = 0$, we find $\delta N = 0$ or $N_L = N_R$. Thus, even if $\Delta$ bifurcates, it is possible that no change in the macroscopic state is seen.

The bifurcation of $\Delta$ indicates formation of a new phase; the coupling constants $g_j$ and the corresponding modes $q_j$ become strongly correlated. This is exemplified in Fig. 2(b) in a simple case of two equally large partitions with two different couplings. The partitions have distinctly different dynamical behaviors and, in the stationary state, the molecules are found mostly in the right well for one partition and in the left well for the other. Such a correlation may be observed even if all the couplings are different.

## 4.3 Change to thermalization or reaction rate

To conclude the analysis of the rate theory approach, let us consider the dynamics of thermalization, i.e., reaction kinematics. To this end, consider an experiment where every molecule is initialized in the left well ("reactant") and, at a time $t$ later, the percentage of the molecules in the right well ("product") is measured. The thermalization (or reaction) rate is initially controlled by $\Gamma_{LR}$ alone. Averaging over Eq. (9), this initial rate is modified from $\Gamma^0_{LR}$ by a factor

$$\langle r_{LR} \rangle \approx \exp\left( -\alpha \frac{\langle g \rangle^2}{\langle g^2 \rangle} + \frac{\alpha^2}{2} \frac{\langle g \rangle^2}{\langle g^2 \rangle} \left[ 1 - \frac{\langle g \rangle^2}{\langle g^2 \rangle} \right] \right), \tag{14}$$

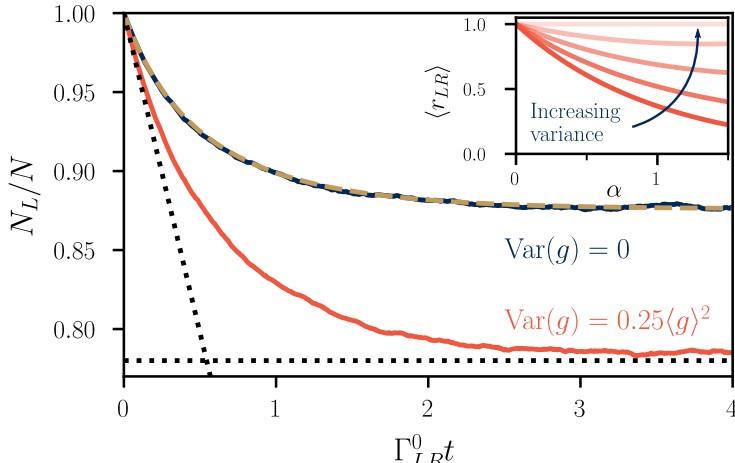

Figure 3: Kinematics for different Gaussian distributions of coupling constants $g_j$. The other parameters are as in Fig. 2(b). The solid lines are obtained by the Markov chain simulation of 1000 molecules averaged over 100 realizations, the dashed line by evaluating Eq. (2), while the dotted black lines represent both the stationary state [Eqs. (11)–(12)] and the initial behaviour [Eq. (14)]. Inset: The rate modification in the initial state $N_L = N$ [Eq. (14)] as a function of $\alpha$. From bottom to top, the different lines are obtained by setting $\langle g \rangle^2 = c \langle g^2 \rangle$ with $c = 1, 0.75, 0.5, 0.25, 0$.

in the leading order of $1/N$ and to the second order in the cumulant expansion. This result, derived in Appendix B.1, is independent of the bias $\sigma$. In the inset of Fig. 3, we show the effect of $\alpha$ and the variance of the coupling distribution. The effective transition rate becomes always slower due to the presence of the cavity — the cavity is an inhibitor — and the effect is stronger the less varied the couplings are.

The agreement between our theoretical approach and the Markov chain simulation is shown in Fig. 3. For equal couplings, $\text{Var}(g) = 0$, we use the macroscopic rate equation (2) directly and the results are nearly indistinguishable. Our solutions for the stationary and transient behavior also agree well with the simulation.

# 5 Phase transition from a quantum mechanical model

Previous section describes how the cavity-induced interaction between molecules leads to a bifurcation in the rate equation model, resembling a phase transition. This phenomenon has some similarities to the superradiant phase transition in the Dicke model, originally discussed by Hepp and Lieb [52]. Namely, a collection of electric dipoles interacting with the electromagnetic vacuum field can be exactly mapped to an Ising model that exhibits a ferromagnetic phase transition. However, while these two phase transitions have some similarities, they are not the same as they exhibit a different set of parameters.

Consider $N$ two-state systems having a dipole operator proportional to $\sigma_z$. Let us assume that there is no energy difference between the two states. A conventional (low-energy) quantum electrodynamical Hamiltonian to describe the interaction of these dipoles with a cavity or vacuum field reads as ($\hbar = 1$) [43]

$$H = \sum_\alpha \omega_\alpha a_\alpha^\dagger a_\alpha + \sum_\alpha \sum_{j=1}^{N} (g_{j\alpha} a_\alpha + g_{j\alpha}^* a_\alpha^\dagger) \sigma_z^j. \tag{15}$$

Here, $\omega_\alpha$ is the eigenfrequency of the cavity mode $\alpha$ to which photons are created by the

operator $a_\alpha^\dagger$ and from which they are annihilated by $a_\alpha$. The light-matter coupling between these cavity modes and a two-level system $j$ are given by $g_{j\alpha}$.

A fully-connected Ising Hamiltonian is now found by applying a unitary transform with $U = \exp\left[\sum_{j\alpha}(1/\omega_\alpha)(g_{j\alpha}^* a_\alpha^\dagger - g_{j\alpha} a_\alpha)\sigma_z^j\right]$, that is,

$$H' = UHU^\dagger = \sum_\alpha \omega_\alpha a_\alpha^\dagger a_\alpha - \sum_\alpha \frac{1}{\omega_\alpha}\left|\sum_{j=1}^N g_{j\alpha}\sigma_z^j\right|^2 \to \sum_\alpha \left(\omega_\alpha a_\alpha^\dagger a_\alpha - \sum_{j\neq k}^N \frac{g_{j\alpha}^* g_{k\alpha}}{\omega_\alpha}\sigma_z^j\sigma_z^k\right), \quad (16)$$

where we have neglected a constant energy shift in the last step.

The spontaneous symmetry breaking associated with the Ising model corresponds to the alignment of the electric dipoles $\sigma_z^j$ and the spontaneous polarization of the cavity field. For the sake of a simple argument, consider only a single relevant cavity mode $\alpha$ to which all the two-level systems couple equally with strength $g$. The effective Ising Hamiltonian becomes $-\frac{g^2}{\omega_\alpha}\sum_{j\neq k}\sigma_z^j\sigma_z^k$. If embedded in a bath of temperature $T$, the Ising model's phase transition happens when $T < T_c = (N-1)\frac{g^2}{k_B \omega_\alpha}$, which spontaneously aligns the two-level systems to the same state [53].

Let us compare this to the critical temperature (13) obtained within the rate theory analysis. It is analogous in that both contain spontaneous symmetry breaking but, in the case of a semiclassical stochastic process, the potential energy surface increases the critical temperature by a factor of $2E_b/\omega_b$ compared to the Ising model. In the Ising model literature, the two-level systems describe the spins of a ferromagnet and their alignment corresponds to a finite magnetization. Here, however, the electric dipoles are aligned. As a direct consequence, the electric field of the cavity mode, proportional to $\langle a_\alpha + a_\alpha^\dagger\rangle$, breaks the symmetry as well, obtains a finite expectation value, and the vacuum becomes polarized.

The existence of an equilibrium phase transition in QED involving such a superradiant or a photon condensate state has been debated over the years. Often, this is formulated in terms of no-go theorems which show that such transition cannot happen due to gauge invariance which is broken when the self-interaction term is neglected [54]. However, these no-go theorems rely on strong assumptions such as that the molecules are located in a region much smaller than the wavelength of the cavity field [55] and that there are no direct intermolecular interactions [38,56]. In the context of polaritonic chemistry, at least, such assumptions do not respect physical reality of the experiments. Furthermore, it has been argued that the direct Coulomb interactions between the dipoles exactly cancel the self-interaction term so that the one is left with the Dicke model as in Eq. (15) [38].

The spontaneous symmetry breaking is not a resonant effect, caused by tuning the cavity eigenfrequencies $\omega_\alpha$ which is often associated with polaritonics. However, it is intrinsically a collective effect. One cannot find this phase transition in a model consisting of a single two-level system.

# 6 Conclusion

To summarize, we have shown how a cavity-mediated interaction between molecules can cause a collective change in their stochastic behavior. The cavity may cause a slowdown of the thermalization rate of the full system and a bifurcation in the stationary state. The cavity effect depends on the number of molecules and the distribution of the light-matter couplings. The phase transition can even show up as a phase separation without altering the macroscopic state. The semiclassical model based on classical rate theory that underlies these results can be understood as an effective dipole-dipole interaction mediated by the electromagnetic vacuum

field. This we illustrate by showing that a quantum mechanical light-matter Hamiltonian can be mapped to a fully-connected Ising model of interacting dipoles in a special case.

In polaritonic chemistry, our approach is straightforward to extend to studies of multiple cavity modes, molecular symmetries [57], and cavity-induced reaction selectivity [19]. Thermodynamic extensions should also open a fruitful avenue of study: Here, the light-matter system forms a canonical ensemble but, instead of connecting to classical rate theory, one may investigate its thermodynamic properties. The phase transition should have associated thermodynamic signatures, such as latent heat, whose magnitude and significance is unknown. Since we expect that, experimentally, the molecules and their dipoles are highly disordered, these thermodynamic signatures might provide the most practical observable to measure while the reaction rates or equilibrium concentrations are unaffected. Using the approach of this paper, it seems possible to consider grand canonical ensembles to describe a much wider variety of chemical systems. Furthermore, it is not clear to us how to incorporate cavity-induced phase transition physics into conventional rate theories used in chemistry, e.g., transition-state theory.

Our work indirectly raises the question whether it is possible to replace the phenomenological description of the light-matter coupling by a microscopic QED description. That is, one might be able to derive an expression for the effective dipole-dipole interaction mediated by the cavity vacuum field in terms of the physical parameters, such as the cavity size, dielectric constant within the cavity, and fine-structure constant. It would elucidate the physical mechanism of the cavity-mediated interaction as well as explain the critical temperature (13) with only the physical parameters of the system.

Due to the ubiquitous nature of light-matter coupling model used here, our results can be used to describe other systems such as electric circuits and cold atoms in optical traps. In these systems the disorder of the effective dipoles can be controlled to a much greater degree than in polaritonic chemistry experiments. This allows for direct observation of the phase transition and its effect on thermalization.

## Acknowledgments

We thank Mark Dykman, Andreas Wacker, Dmitry Morozov, Jussi Toppari, and Gerrit Groenhof for the insightful discussions that have contributed to this work.

**Funding information**   This project was supported by the Magnus Ehrnrooth foundation and the Academy of Finland (project numbers 317118 and 321982).

## A   Calculation of the state-dependent potential energy

Here, we present how we calculate the potential energies of the classical states, starting from the total potential

$$V_{\text{tot}}(x, q_1, \ldots, q_N) = \sum_{i=1}^{N} V(q_i) + \frac{1}{2}\omega_c^2 x^2 + \sum_{i=1}^{N} d_i(q_i)x \,, \qquad (A.1)$$

where $V(q_i)$ is a bistable potential and $d_i(q_i)$ characterizes the dipole moment of the $i$th molecule projected on the polarization vector of the cavity mode. We point out that the following calculation is very similar to the Caldeira–Leggett model except that the typical bath of harmonic oscillators is replaced by a single mode. Consequently, instead of dissipation,

integration over the cavity mode $x$ leads to effective interaction between the molecular coordinates $q_i$.

First, we solve the stationary points dictated by the conditions $\partial_x V_{\text{tot}} = 0 = \partial_{q_i} V_{\text{tot}}$. We work here in the harmonic approximation, assuming that the coupling to the cavity quadrature $x$ induces a small change in the positions of the stationary points $q_i^*$ which are the minimum points of the potential fulfilling $\partial_{q_i} V(q_i)\big|_{q_i = q_i^*} = 0$. We choose the coordinate system so that $q_i^* = \pm a/2$ without the presence of the cavity field. Thus, $a$ is the distance between the (local) minimum points and the molecular state can be encoded into $s_i = \text{sign}(q_i) = \pm 1$. To characterize the displacement of the minimum points, we introduce a variable $\delta q_i = q_i - \frac{a}{2} s_i$. We can now write the potential in the harmonic approximation as

$$V(q_i) \approx -\frac{\sigma}{2} s_i + \frac{1}{2} \omega_i^2 \delta q_i^2 , \tag{A.2}$$

where $\sigma$ represents the energy difference between the left and right well and $\omega_i^2$ is a shorthand notation for

$$\omega_i^2 \equiv \omega^2(s_i) = \frac{\omega_R^2 + \omega_L^2}{2} + \frac{\omega_R^2 - \omega_L^2}{2} s_i , \tag{A.3}$$

which alternates between $\omega_L^2$ and $\omega_R^2$ depending on the molecular state. Similarly, we assume that the dipole moment can be linearized near $q_i = \pm a/2$

$$d_i(q_i) \approx \lambda_i^2(s_i) \left[ \delta q_i + \frac{q_0}{2} s_i \right], \tag{A.4}$$

where $\lambda_i^2(s_i)$ describes the (possibly state-dependent) strength of the light-matter coupling and $q_0$ additionally characterizes the difference in the dipole moment between right and left well. The coupling constants $\lambda_i^2(s_i)$ can also vary from molecule to molecule, for instance, due to disorder in the polarization vectors and positions. Here, we choose $\lambda_i(s_i)$ to have the units of frequencies $\omega_i$, but it should be noted that $\lambda_i^2(s_i)$ can be negative. The results of the main paper are obtained by replacing $q_0 = a$ and $\lambda_i^2(s_i) \to \sqrt{\omega_m \omega_c} g_i$ where $g_i$ is independent of the molecular state. The presence of the square root term is due to the choice of coordinates $q_i$ and $x$ so that, for a harmonic potential $V(q_i)$, we retain the light-matter interaction with constants $g_i$ as $g_i(a^\dagger + a)(b_i^\dagger + b_i)$ after quantization as in the Dicke model.

As the derivative of the sign function $s_i = \text{sign}(q_i)$ vanishes everywhere except at $q_i = 0$, and $q_i$ obtains values only near $\pm a/2$ by assumption, we find

$$0 = \frac{\partial V_{\text{tot}}}{\partial x} = \omega_c^2 x + \sum_i \lambda_i^2(s_i) \left[ \delta q_i + \frac{q_0}{2} s_i \right], \tag{A.5a}$$

$$0 = \frac{\partial V_{\text{tot}}}{\partial q_i} = \omega_i^2 \delta q_i + \lambda_i^2(s_i) x . \tag{A.5b}$$

This set of equations can be solved by noting that Eq. (A.5a) depends on $Q = \sum_i \lambda_i^2(s_i) \delta q_i$ and, at the same time, we can derive from Eq. (A.5b) an equation

$$0 = Q + \sum_i \frac{\lambda_i^4(s_i)}{\omega_i^2} x , \tag{A.6}$$

for the collective variable $Q$. Inserting the solution of $Q$ in terms of $x$ to Eq. (A.5a) gives

$$x = -\frac{\frac{q_0}{2} \sum_i s_i \lambda_i^2(s_i)}{\omega_c^2 - \sum_i \frac{\lambda_i^4(s_i)}{\omega_i^2}} = -\frac{\omega_m^2}{\sqrt{\langle \lambda^4 \rangle_s}} P \Delta \frac{q_0}{2} . \tag{A.7}$$

In the latter equality, we introduce the useful notations from the main text — which we gather here for convenience

$$\omega_m^2 = \frac{\omega_R^2 + \omega_L^2}{2}, \quad \langle \lambda^4 \rangle_s = \frac{\sum_i \lambda_i^4(s_i)}{N}, \quad P = \frac{N \langle \lambda^4 \rangle_s}{\omega_c^2 \omega_m^2 - \sum_i \frac{\omega_m^2}{\omega_i^2} \lambda_i^4(s_i)}, \quad \Delta = \frac{1}{N} \sum_i \frac{\lambda_i^2(s_i)}{\sqrt{\langle \lambda^4 \rangle_s}} s_i .$$

$$(A.8)$$

Though, here, the average coupling $\langle \lambda^4 \rangle_s$ still depends on the states of the molecules which is indicated by the subscript $s$. The solution of $x$ allows for the solutions of individual $\delta q_i$ which are simply $\delta q_i = -\frac{\lambda_i^2(s_i)}{\omega_i^2} x$. Note that the solution is consistent with the harmonic approximation as long as the dimensionless constant $P$ is well below unity and $\frac{|\omega_L^2 - \omega_R^2|}{\omega_L^2 + \omega_R^2} \ll 1 - P$.

Finding the potential minima is now straightforward. We insert the obtained local minimum points into the expression of total potential $V_{\text{tot}}$. The potential energies found in this way still depend on the state of the system in the sense that the variables $s_i = \text{sign}(q_i)$ are not fixed. Within the harmonic approximation it holds that $|\delta q_i| < a/2$ and it is clear that the signs of $q_i$ indicate whether the system is in the left ($q_i < 0, s_i = -1$) or right ($q_i > 0, s_i = +1$) well. We find

$$\begin{aligned} V_{\text{tot}} &= -\frac{\sigma}{2} \delta N + \sum_{i=1}^N \frac{1}{2} \omega_i^2 \delta q_i^2 + \frac{1}{2} \omega_c^2 x^2 + \sum_{i=1}^N \lambda_i^2 x \left[ \delta q_i + \frac{q_0}{2} s_i \right] \\ &= -\frac{\sigma}{2} \delta N + \frac{1}{2} \left( \omega_c^2 - \sum_i \frac{\lambda_i^4}{\omega_i^2} \right) x^2 + \frac{q_0}{2} x \sum_{i=1}^N \lambda_i^2 s_i \\ &= -\frac{\sigma}{2} \delta N - \frac{N}{2} \omega_m^2 \left( \frac{q_0}{2} \right)^2 P \Delta^2 , \end{aligned} \quad (A.9)$$

which matches with the result of the main text when $E_b = \frac{1}{2} \omega_m^2 \left( \frac{q_0}{2} \right)^2$ is identified in the latter term. Since $E_b$ has the dimension of energy, we call it the effective potential barrier.

In this article, for the most part, we focus on dipole moment functions $d_i(q_i)$ that can be written as a linear function over the whole range of $q_i$. Physically, this means that both molecular states are equally coupled to the cavity. The condition is easily relaxed to investigate a multitude of possible polaritonic systems — we briefly discuss one example in Sec. F. In terms of the model at hand, it means that $d_i(q_i) = \lambda_i^2 q_i$ where $\lambda_i^2$ is a constant for each molecule $i$. Thus, the average $\langle \lambda^4 \rangle$ is state independent and, as seen in the next section, $P$ can be regarded as a constant in the lowest order of the light–matter coupling. In addition, if also $\omega_L = \omega_R$, $E_b$ simply describes the potential energy $V(q=0)$ within the harmonic approximation since $q_0 = a$.

## B  Detailed balance and the transition rate modification

The detailed balance states that the transition rates must obey

$$\frac{\Gamma_{LR}^j}{\Gamma_{RL}^j} = \exp\left\{ \beta \left[ V_{\text{tot}}(\dots, s_{j-1}, -1, s_{j+1}, \dots) - V_{\text{tot}}(\dots, s_{j-1}, +1, s_{j+1}, \dots) \right] \right\} = \exp(\beta \delta V_{\text{tot}}), \quad (B.1)$$

in order for the system to thermalize to the Boltzmann distribution. We assume here that the light-matter coupling is state independent and hence $\lambda_i^2(s_i) = \lambda_i^2$.

First, we calculate the difference in $P = P(s_1, \dots, s_N)$ in Eq. (A.8) due to one molecular state change. It is useful to denote $K = \frac{1}{N} \sum_i \frac{\lambda_i^4}{\langle \lambda^4 \rangle} \frac{\omega_m^2}{\omega_i}$. Note that $K = K(s_1, \dots, s_N)$ but $K = 1$ if

$\omega_L = \omega_R$. Thus, we formally can expand $P$ around $K = 1$ which leads to

$$P = \sum_{n=0}^{\infty} \left( \frac{N \langle \lambda^4 \rangle}{\omega_c^2 \omega_m^2 - N \langle \lambda^4 \rangle} \right)^{n+1} (K-1)^n \equiv \sum_{n=0}^{\infty} P_0^{n+1}(K-1)^n. \tag{B.2}$$

Since we assume that $N \langle \lambda^4 \rangle / (\omega_c^2 \omega_m^2)$ is small and $P_0 \ll 1$, $P \approx P_0 + P_0^2(K-1)$ to a good approximation. The higher powers of $K$ also scale in the powers of $1/N$. Finding the difference is now straightforward and results in

$$\delta P_j = P(\dots, s_{j-1}, -1, s_{j+1}, \dots) - P(\dots, s_{j-1}, +1, s_{j+1}, \dots)$$
$$= \frac{1}{N} P_0^2 \frac{\lambda_j^4}{\langle \lambda^4 \rangle} \left( \frac{\omega_m^2}{\omega_L^2} - \frac{\omega_m^2}{\omega_R^2} \right) + \mathcal{O}(P_0^3/N^2). \tag{B.3}$$

We neglect the second order contribution of $P_0$ in our analysis and, thus, the state dependency of $P \approx P_0$ as $\delta P_j \approx 0$.

Since $P$ can be regarded as a constant in the calculation of the difference in potential energies, the calculation simplifies greatly and we obtain

$$\delta V_{\text{tot}} = \sigma - N E_b P \left[ \left( \frac{-\frac{\lambda_j^2}{\sqrt{\langle \lambda^4 \rangle}} + \sum_{i \neq j} \frac{\lambda_i^2}{\sqrt{\langle \lambda^4 \rangle}} s_i}{N} \right)^2 - \left( \frac{\frac{\lambda_j^2}{\sqrt{\langle \lambda^4 \rangle}} + \sum_{i \neq j} \frac{\lambda_i^2}{\sqrt{\langle \lambda^4 \rangle}} s_i}{N} \right)^2 \right]$$
$$= \sigma + 4 P E_b \frac{\lambda_j^2}{\sqrt{\langle \lambda^4 \rangle}} \frac{\sum_{i \neq j} \frac{\lambda_i^2}{\sqrt{\langle \lambda^4 \rangle}} s_i}{N}. \tag{B.4}$$

However, we still have to connect this to what would be observed in experiments which deal with macroscopic numbers of $N$.

With the value of $\delta V_{\text{tot}}$, the detailed balance states that

$$\frac{\Gamma_{LR}^j}{\Gamma_{RL}^j} = \frac{\Gamma_{LR}^0}{\Gamma_{RL}^0} \frac{r_{LR}^j}{r_{RL}^j} = e^{\beta \sigma} \exp\left( 4 P \beta E_b \frac{\lambda_j^2}{\sqrt{\langle \lambda^4 \rangle}} \frac{\sum_{i \neq j} \frac{\lambda_i^2}{\sqrt{\langle \lambda^4 \rangle}} s_i}{N} \right). \tag{B.5}$$

Here, one can readily identify the ratio of the rate modification factors which is the latter exponent.

Now, we can appeal to symmetry: if we interchange left and right everywhere in the potential $V(q)$, the rate modifications should remain the same. Thus,

$$r_{LR}^j(\{q_i\}; \sigma, \omega_L^2, \omega_R^2) = r_{RL}^j(\{-q_i\}; -\sigma, \omega_R^2, \omega_L^2), \tag{B.6}$$

where $\{q_i\}$ refers to the set of all $q_i$'s. One can confirm that $P$ remains invariant in this transformation (the relevant quantity in $P$ is $\sum_i \lambda_i^4 \frac{\omega_m^2}{\omega_i^2} = \sum_i \frac{\lambda_i^4}{1 + s_i(\omega_R^2 - \omega_L^2)/(\omega_R^2 + \omega_L^2)}$). Following the line of deduction as in the main text, we can thus write

$$r_{LR}^j = \exp\left( 2 P \beta E_b \frac{\lambda_j^2}{\sqrt{\langle \lambda^4 \rangle}} \frac{\sum_{i \neq j} \frac{\lambda_i^2}{\sqrt{\langle \lambda^4 \rangle}} s_i}{N} \right), \tag{B.7}$$

and $r_{RL}^j = 1/r_{LR}^j$. We may also replace the sum in this expression with $\Delta$ as the error made is of the order of $1/N$.

## B.1 Cumulant expansion

Next, we treat the coupling constants $\lambda_j^2$ as independent random variables — instead of considering them to have some fixed values — and average the detailed balance relation over all realizations. Furthermore, we assume that the molecular state $s_j$ is independent of $\lambda_j^2$ which does not necessarily hold for a dynamical system but may be assumed to hold for the initial state, for instance. The average of the exponent gives rise to the cumulant expansion. To calculate the cumulants, we first calculate the raw moments of the term in the exponent. Taking only the leading order in $1/N$, we find

$$\left\langle \left[ \frac{\lambda_j^2}{\sqrt{\langle \lambda^4 \rangle}} \frac{\sum_{i \neq j} \frac{\lambda_i^2}{\sqrt{\langle \lambda^4 \rangle}} s_i}{N} \right]^n \right\rangle \approx \left\langle \left( \frac{\lambda^2}{\sqrt{\langle \lambda^4 \rangle}} \right)^n \right\rangle \left[ \frac{\langle \lambda^2 \rangle}{\sqrt{\langle \lambda^4 \rangle}} \frac{\delta N}{N} \right]^n. \tag{B.8}$$

Since all cumulants may be expressed using only raw moments in a specific polynomial manner [1], we can deduce from this result that the $n$th cumulant $K_n$ of the term in the exponent of $r_{LR}^j$ is

$$K_n = \left[ 2P\beta E_b \frac{\langle \lambda^2 \rangle}{\sqrt{\langle \lambda^4 \rangle}} \frac{\delta N}{N} \right]^n K_n \left( \lambda^2 / \sqrt{\langle \lambda^4 \rangle} \right). \tag{B.9}$$

That is, we have related the cumulants of the sum to the cumulant of its individual elements. Using the cumulant expansion to second order, we have that

$$\ln \langle r_{LR} \rangle \approx 2P\beta E_b \frac{\langle \lambda^2 \rangle^2}{\langle \lambda^4 \rangle} \frac{\delta N}{N} + \frac{1}{2} (2P\beta E_b)^2 \frac{\langle \lambda^2 \rangle^2}{\langle \lambda^4 \rangle} \left[ 1 - \frac{\langle \lambda^2 \rangle^2}{\langle \lambda^4 \rangle} \right] \left( \frac{\delta N}{N} \right)^2. \tag{B.10}$$

Here, the first term follow from the average and the second from the variance. We emphasize that the approximation is valid when $N \gg 1$. The cumulant expansion is otherwise exact if the distribution of $\lambda_j^2$'s is Gaussian; only the first two cumulants may be finite for a Gaussian distribution.

## C  Derivation of the self-consistency equation

One can derive a self-consistency equation for $\Delta$ in the stationary state. Let us assume that, even though there are $N$ systems, there are $M < N$ different values for the coupling constants $\lambda_j^2$. Let us denote $c_j = \frac{\lambda_j^2}{\sqrt{\langle \lambda^4 \rangle}}$ and $\alpha = 2P\beta E_b$ here, for brevity. We divide $N$ into $M$ subsystems or partitions so that $N = \sum_{k=1}^M N^k$. Each subsystem $k \in \{1, 2, \dots M\}$ has $N_k$ systems which all share the same coupling constant $c_j = c_k$. Then, $\Delta = \frac{1}{N} \sum_k c_k (N^k - 2N_L^k)$ where $N_L^k$ is the number of systems in the left well. Following the main text, we write for each subsystem $k$ the macroscopic equation

$$\frac{d}{dt} N_L^k = -\Gamma_{LR}^k N_L^k + \Gamma_{RL}^k (N^k - N_L^k). \tag{C.1}$$

We assume that each subsystem has a stationary value. We may then solve, using Eq. (B.5),

$$\frac{N_L^k}{N^k} = \frac{1}{1 + \exp(\beta \sigma + 2\alpha c_k \Delta)}. \tag{C.2}$$

Inserting this solution to the definition of $\Delta$ and $\delta N$ leads to

$$\frac{\delta N}{N} = 1 - 2\frac{\sum_k N_L^k}{N} = \sum_k \frac{N^k}{N}\tanh\left(\beta\frac{\sigma}{2} + \alpha c_k\Delta\right) \equiv \left\langle\tanh\left(\beta\frac{\sigma}{2} + \alpha c\Delta\right)\right\rangle, \qquad (C.3)$$

$$\Delta = \sum_k \frac{c_k}{N}(N^k - 2N_L^k) = \sum_k \frac{N^k}{N}c_k\tanh\left(\beta\frac{\sigma}{2} + \alpha c_k\Delta\right) \equiv \left\langle c\tanh\left(\beta\frac{\sigma}{2} + \alpha c\Delta\right)\right\rangle. \qquad (C.4)$$

In both equations, we identified the structure of $\sum_k(N^k/N)f(c_k)$ as an average over the distribution of $c$'s which follows from the fact that there are exactly $N^k$ values of $c_k$. In the limit $N \to \infty$, we may construct a continuous distribution of the coupling constants which we use here as an approximation for systems of finite size.

It should be noted that we treat $\Delta$ and $\delta N$ as numbers, not stochastic variables, even though the underlying stochastic processes determine their realizations and the possible values of $\Delta$ are constrained by the distribution of the coupling constants.

In this derivation, we have neglected the possible variation of $\alpha$ with the macroscopic state. This arises from the variation of $P$, as discussed in Appendix B. In principle, this is remedied by including an equation for $\alpha = \alpha(\Delta)$ and solving all three equations self consistently. It should be noted that the variation of $P$ with the macroscopic state is a second-order effect in $P$ and, thus, one should also calculate $\delta V_{\text{tot}}$ to the same order.

## D  Relation to the full master equation of the $N$ molecule system

If all the coupling constants $g_j$ are equal, the potential difference $\delta V_{\text{tot}}$ is independent of the system index $j$. In this case, the full $N$ molecule system is readily described by the probability $P(N_L)$ to find $N_L$ systems in the left well. The master equation for these probabilities reads as

$$\frac{\mathrm{d}}{\mathrm{d}t}P(N_L) = \mu_{N_L+1}P(N_L+1) + \nu_{N_L-1}P(N_L-1) - (\mu_{N_L} + \nu_{N_L})P(N_L), \qquad (D.1)$$

where $\mu_M = M\Gamma_{LR}^0 r_{LR}(M)$ describes the removal rate of particles from the left well to the right well and $\nu_M = (N-M)\Gamma_{RL}^0 r_{RL}(M)$ the inverse process. Now, the expectation value of $N_L$ is given by $\langle N_L\rangle = \sum_{N_L} N_L P(N_L)$. Using this definition, one can derive from the master equation an equation for the average as

$$\frac{\mathrm{d}}{\mathrm{d}t}\langle N_L\rangle = -\Gamma_{LR}^0\langle r_{LR}N_L\rangle + \Gamma_{RL}^0\langle r_{RL}(N-N_L)\rangle. \qquad (D.2)$$

This matches exactly to the macroscopic rate equation in Eq. (2) if $\langle r_{LR}N_L\rangle = \langle r_{LR}\rangle\langle N_L\rangle$ and $\langle r_{RL}(N-N_L)\rangle = \langle r_{RL}\rangle(N - \langle N_L\rangle)$, and the resulting rate modification factors may be understood as a function of the mean value of $N_L$ only. Due to the non-linearity of the rate modifications, this can be used as a valid approximation only when the fluctuations of $N_L$ are small compared to some characteristic scale given by the rate modifications.

This assumption can be checked numerically. First, we note that the transition rates are independent of time. The master equation (D.1) can thus be written in a matrix form as $\partial_t\vec{P} = W\vec{P}$ where $W$ is independent of time. Given an initial state $\vec{P}(t=0)$, we formally have $\vec{P}(t) = e^{Wt}\vec{P}(t=0)$. The numerical problem is then to calculate the matrix exponential $e^{Wt}$. Fig. 4 shows that the macroscopic rate equation and the master equation produces the same mean behavior for $N_L$ as $N$ is large enough.

We note that even though the rate equations are state dependent and consequently non-linear, the macroscopic behaviour fits well to the typical exponential behaviour

$$N_L(t) = N_L(\infty) + [N_L(0) - N_L(\infty)]e^{-\gamma t}, \qquad (D.3)$$

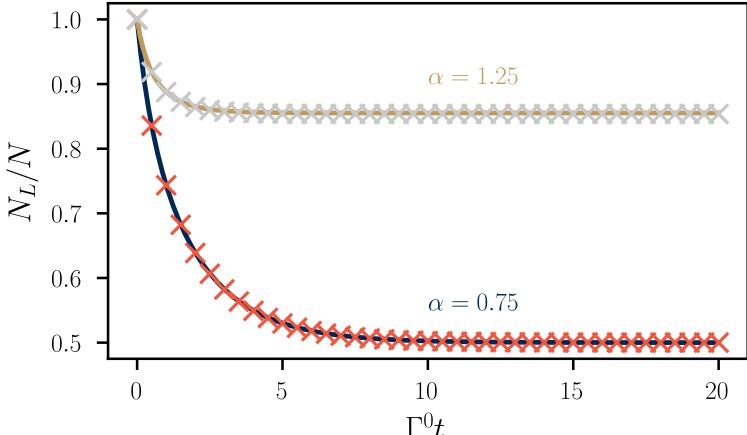

Figure 4: Comparison between the macroscopic rate equation (solid lines) and master equation (crosses). Here, $\Gamma^0_{LR} = \Gamma^0_{RL} = \Gamma^0$ and $N = 1000$.

with a suitable fitting constant $\gamma$. This observation seems to hold even when the variation of couplings is included which is possible by the numerical method presented in the next section.

## E  Numerical Markov chain approach

Here, we shortly describe the Markov chain algorithm we use in this work. The main task is to simulate a set of rate equations

$$\frac{d}{dt}p_j = -\Gamma^j_{LR}p_j + \Gamma^j_{RL}(1 - p_j), \tag{E.1}$$

where $\Gamma^j_{\mu\nu}$ depend on the state of the system. In our formulation, these states are $s_j = \text{sign}(q_j)$ so that $s_j(t) \in \{-1, +1\}$. The rate equation describes that in a small time step $\Delta t$, the probability to change the system from $s_j = -1$ to $s_j = +1$ is given by $\Gamma^j_{LR}\Delta t$ while the converse process has the probability $\Gamma^j_{RL}\Delta t$. The algorithm is as follows:

1. Initialize the states $\{s_j\}$ at time $t$ if necessary.

2. Draw $N$ random numbers $u_j \sim \text{Uniform}(0, 1)$.

3. For each $j \in \{1, 2, \dots N\}$ calculate $A_j = \Gamma^j_{LR}\Delta t$ if $s_j = -1$ and $A_j = \Gamma^j_{RL}\Delta t$ if $s_j = +1$ using the system state at time $t$.

4. For each $j \in \{1, 2, \dots N\}$ compare $u_j$ and $A_j$. If $u_j < A_j$, set the system state at time $t + \Delta t$ to be $s_j(t + \Delta t) = -s_j(t)$. Otherwise, set $s_j(t + \Delta t) = s_j(t)$.

5. Return to step 1 with $t \to t + \Delta t$.

Thus, when an appropriate amount of time steps is taken, we have a list of system states $s_j(t)$ at the chosen time points. These states can be used to calculate e.g. $\Delta$ or $\delta N$. This is exemplified in Fig. 5.

Practically, we choose $N$ to be large enough so that there are no spurious $1/N$ effects and correspondence with the master equation approach holds. The time step we choose so that $\Gamma^0_{\mu\nu}\Delta t$ is below or of the order of 1 percent. The initial state is chosen according to how many systems are wanted to be found in the left well and then the states are shuffled.

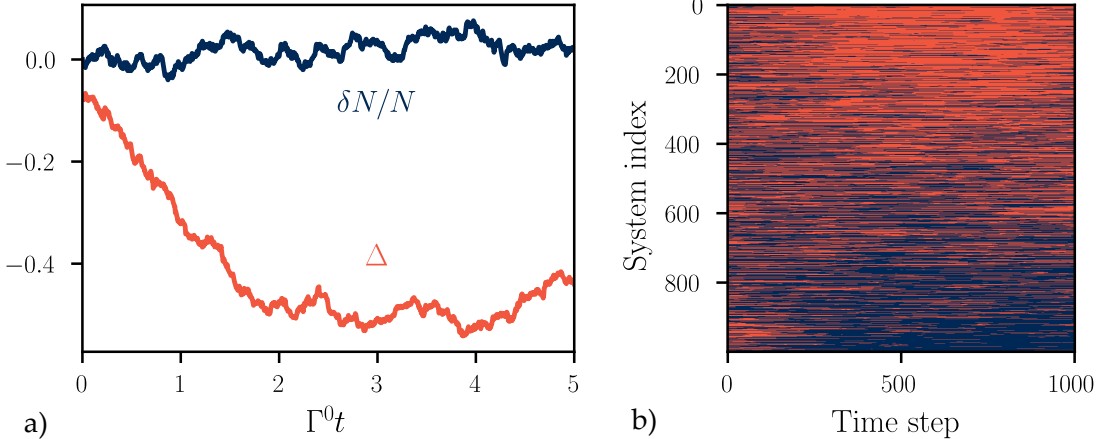

Figure 5: A single realization of the Markov chain simulation for $N = 1000$ systems and 1000 time steps ($\Gamma^0 \Delta t = 0.01$). Other parameters are $\alpha = 1.3, \sigma = 0$, and $\Gamma^0_{LR} = \Gamma^0_{RL} = \Gamma^0$. Here, the coupling constants are sampled from a Gaussian distribution with zero mean. a) The time evolution of the quantities $\delta N$ and $\Delta$. As discussed in the main text, $\Delta$ has a solution other than zero when $\alpha > 1$ while $\delta N$ fluctuates around zero. b) As a visual representation, the states "left" and "right" are here represented by blue and orange, respectively, for each time step. Here, the different systems are indexed in the ascending order in the coupling constant (i.e., $g_1 \leq g_j \leq g_N$). The correlation becomes visible; the systems with largest values of coupling are predominantly found in the left well after a transient period.

## F  Effect of nonlinear dipole moment in a symmetric potential

We shortly describe how the physics changes when the assumption of equal light-matter coupling strength in both wells is lifted. Now, we have to specify two couplings $\lambda_i^2(s_i = -1) = \lambda_{iL}^2$ and $\lambda_i^2(s_i = +1) = \lambda_{iR}^2$ for each molecule in Eq. (A.4) based on its state. Furthermore, the average coupling on different states may differ, that is $\left\langle \lambda_L^2 \right\rangle \neq \left\langle \lambda_R^2 \right\rangle$ where $\left\langle \lambda_{L/R}^2 \right\rangle = \frac{1}{N} \sum_{i=1}^N \lambda_{i\,L/R}^2$. For simplicity, we assume here a symmetric potential with $\omega_L = \omega_R$ and $\sigma = 0$.

Physically, breaking the (anti)symmetry of the dipole moment leads to a preference or a lower energy on the state that is coupled more strongly to light. This is relevant for engineering state or reaction selectivity by vacuum fields. The bifurcation behavior presented in the main text also changes. Here, we investigate this numerically with the help of the Monte Carlo algorithm presented in the previous section.

The main mathematical difficulty in the case of a state-dependent light-matter coupling is that the quantity $P$ is no longer a constant. For a linear dipole moment, this dimensionless parameter describes the strength of the Rabi splitting. Now, assuming relatively small couplings and thus taking only the lowest order in $\sqrt{\langle \lambda^4 \rangle_s}/(\omega_c \omega_m)$, it is useful to write $P$ in terms of an auxiliary constant $\tilde{\lambda}^4$

$$P = \frac{N \left\langle \lambda^4 \right\rangle_s}{\omega_c^2 \omega_m^2 - \sum_i \frac{\omega_m^2}{\omega_i^2} \lambda_i^4} \approx \frac{N \left\langle \lambda^4 \right\rangle_s}{\omega_c^2 \omega_m^2} = \frac{\left\langle \lambda^4 \right\rangle_s}{\tilde{\lambda}^4} \frac{N \tilde{\lambda}^4}{\omega_c^2 \omega_m^2} \equiv \frac{\left\langle \lambda^4 \right\rangle_s}{\tilde{\lambda}^4} \tilde{P}. \tag{F.1}$$

Recall that $\left\langle \lambda^4 \right\rangle_s$ depends on the state of the molecules as well. Now, $\tilde{P}$ can characterize the total coupling strength if we choose, for instance, $\tilde{\lambda}^4 = (\langle \lambda_R^4 \rangle + \langle \lambda_L^4 \rangle)/2$ or simply $\tilde{\lambda}^4 = \left\langle \lambda_{L/R}^4 \right\rangle$.

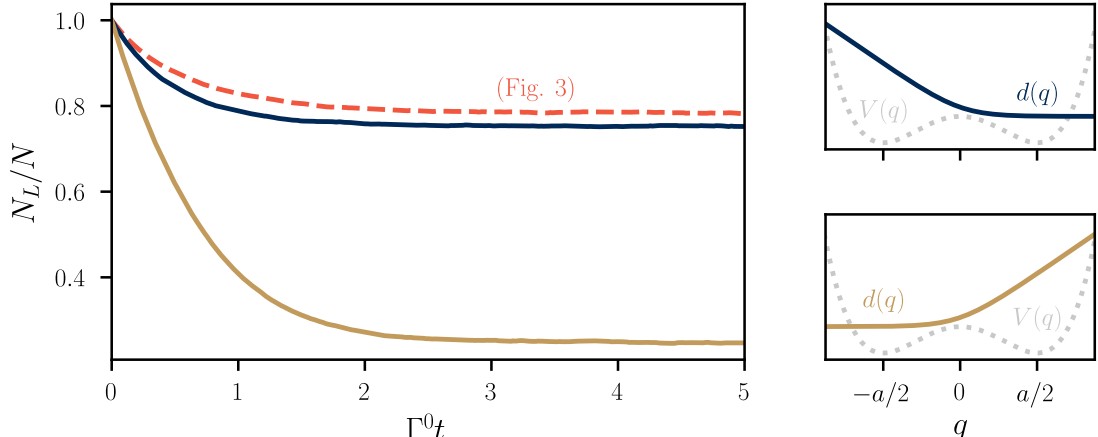

Figure 6: Monte Carlo simulation of a polaritonic system with nonlinear dipole moments, schematized on the right side of the figure. The transition dipole moment vanishes on the right state in the blue line and on the left state in the brown line. The orange line from Fig. 3 is the dashed line here for reference. The simulation uses $N = 1000$ molecules averaged over 100 realizations, and $\Gamma^0_{LR} = \Gamma^0_{RL} = \Gamma^0$.

The calculation of the energy difference between left and right states $\delta V_{\text{tot}}$ follows as in Appendix B. In the lowest order of $1/N$, we find

$$\frac{\Gamma^j_{LR}}{\Gamma^j_{RL}} = e^{\beta \delta V_{\text{tot}}} = \exp\left[ 2\tilde{\alpha} \frac{\langle \lambda^4 \rangle}{\tilde{\lambda}^4} \frac{\lambda^2_{jL} + \lambda^2_{jR}}{2\sqrt{\langle \lambda^4 \rangle}} \left( \Delta - \frac{\lambda^2_{jL} - \lambda^2_{jR}}{2\sqrt{\langle \lambda^4 \rangle}} \Delta^2 \right) \right], \tag{F.2}$$

where $\tilde{\alpha} = 2\tilde{P}\beta E_b$ now functions as a control parameter. It is noteworthy that the nonlinearity of the dipole moment is connected with a $\Delta^2$-term in the energy difference. This term shows that the different light-matter coupling constants generate an effective energy gradient to the molecular states. Furthermore, the collective coupling always vanishes if the dipole moment is fully symmetric so that $\lambda^2_{jR} = -\lambda^2_{jL}$.

Here, we concentrate on a numerical example, even though one could proceed with a similar kind of argumentation as in Sec. B. Let us choose a light-matter coupling that vanishes on one state and is finite on the other. How does the polaritonic system relax, if all the molecules are initially in the left state, and what is the stationary state? To further connect this setup to Fig. 3 in the main text, we choose $\tilde{\lambda}^4$ so that it is the larger one of $\langle \lambda^4_R \rangle$ and $\langle \lambda^4_L \rangle$, choose the couplings from a normal distribution with $\text{Var}(g) = 0.25 \langle g \rangle^2$ (in the description $\lambda^2_i = \sqrt{\omega_m \omega_c} g_i$), and set $\tilde{\alpha} = 1.3$. That is, in the case in which only the molecules on the left state couple to light, the situation is initially the same as in Fig. 3 (orange line).

The result of the numerical simulation is plotted in Fig. 6. As expected, the initial evolution of the macroscopic state is very similar in the case where the left state couples to light as to when both states are coupled to light. The stationary state changes only in a minor way.

Much more interesting is the case where the coupling to light is on the right state only. Initially, there is no light-matter coupling and the polaritonic system starts to thermalize towards an even split between the states. This initial rate is similar to that without any coupling. The stationary state however changes notably as more of the molecules prefer the right state. This again shows the effective attractive interaction mediated by the cavity. Finally, because of symmetry in the potential and the mirror symmetry of the nonlinear dipole moment, the stationary states of these two plotted scenarios are related. One gets from one to the other by simply interchanging the labels for left and right.

Finally, we note that Eq. (F.2) allows solving the stationary states analytically. This is the most straightforward in the case in which there is no variation in the left and right coupling

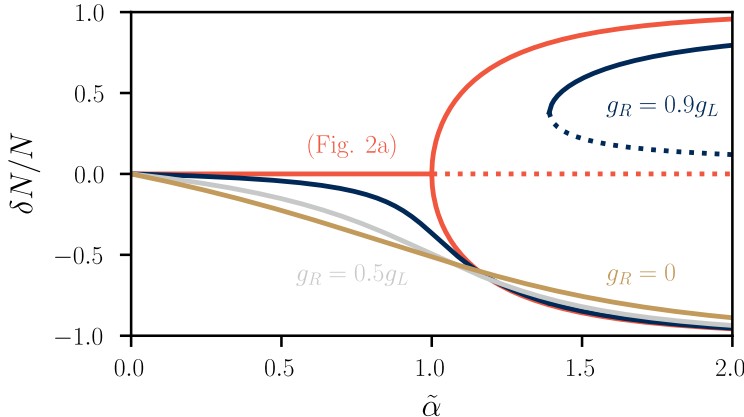

Figure 7: Bifurcation diagram for a symmetric potential without any variation in the left and right light-matter couplings. Here, we fix $\tilde{\lambda}^4 = \lambda_L^4$ within $\tilde{\alpha}$ and the different curves represent different values of $\lambda_R^2$.

constants, that is, $\lambda_{iL}^2 = \lambda_L^2$ and $\lambda_{iR}^2 = \lambda_R^2$. Focusing still on the symmetric potential, the relevant parameter is the ratio $\lambda_R^2/\lambda_L^2 = g_R/g_L$. The effect on the bifurcation diagram, comparing to Fig. 2(a) in the main text, is similar to having a bias in the potential as shown in Fig. 7. That is, the critical value of $\tilde{\alpha}$ at which the bifurcation happens increases as the ratio $g_R/g_L$ deviates from unity. Similar to the effect of bias, there is a measurable effect before the critical value as more molecules are found in the state with the larger coupling.

## G  Realizations of cavity-induced bifurcation

### G.1  Polaritonic chemistry

Let us discuss the validity range of the approach in the main text in the context of polaritonic chemistry. As we derive the rate constant modification from detailed balance, our results disregard quantum effects. That is, the temperature must be above a certain threshold frequency that is related to $\omega_c, \omega_L, \omega_R$ [11]. This limits the systems mostly to vibrational strong coupling where these frequencies are typically of the order of 100 meV and not in the range of several eV as in electronic strong coupling (at room temperature $k_B T \sim 25$ meV). More notably, we treat the molecules as one-dimensional systems and assume a constant bilinear coupling to the (single mode) cavity. We believe that even though this choice neglects many details of molecules used in recent experiments, it gives an important analytical insight which can be improved upon with specific molecular models.

We note that the potential $V(q)$ with $\sigma = 0$ and $\omega_L = \omega_R$ is similar to the ground state potential of the Shin-Metiu model describing proton-coupled electron transfer [58]. Recently, it has been used in other works in polaritonic chemistry [40, 41]. In these works, the typical barrier energy is of the order of 1 eV; although it depends on the exact choice of model parameters. Since the effective potential barrier $E_b$ is typically larger than the barrier energy, our order of magnitude estimate is that $E_b/(k_B T)$ may range from tens to hundreds.

To identify the strong coupling, the Rabi splitting $\Omega$ must be large enough to be observed. This means that $\Omega \gtrsim (\kappa + \gamma)/2$ where $\kappa$ and $\gamma$ refer to the dissipation rates of the cavity and the molecular mode, respectively. We have assumed the collective strong coupling regime where $\Omega/\omega_m \sim 0.1$ leading to $P \sim 0.01$. Thus, all the results are given in the first order of $P$ and higher order corrections are yet to be found. Experimentally, it has been shown that it

is possible to reach the so-called ultrastrong coupling regime in vibrational strong coupling; $\Omega/\omega_m \approx 0.24$ in Ref. [59].

Therefore, the control parameter $\alpha = 2P\beta E_b$ can be of the order of unity in experiments of polaritonic chemistry. For example, if $P = 0.01$ and $\beta E_b = 100$, we have $\alpha = 2$. In the main text, we find that the cavity-induced phase transition happens when $\alpha > 1$ for systems with two metastable states at the same energy.

We emphasize the fact that the coupling constants can differ from molecule to molecule which, somewhat surprisingly, is often neglected in the seminal theoretical works [46]. To further complicate the picture, the couplings may truly be time dependent if the molecules can diffuse within the cavity. However, such time dependent variation is much faster than the typical reaction rates, and thus, we can deal with only the averages of coupling constants. There are at least two types of randomness in the couplings: 1) "orientation disorder" due to the varying directions of transition dipole moments with respect to the cavity polarization vector(s), and 2) "position disorder" due to the varying positions of the molecules within the cavity. These disorders have been shown and studied in several experiments, for instance, in plasmonic picocavities [44] and in optical cavities [45]. In the main text, we find that the average of the coupling constants determines the macroscopic properties of the $N$ molecule polaritonic system, for instance the stationary state and the rate modification. The detailed understanding of these disorders is thus important.

It is speculated in Ref. [12] that the orientation disorder must be negligible in order to see polaritonic chemistry. This indeed corresponds to the rate modification analysis in the main text, as assuming that the transition dipole moments are distributed isotropically would lead to the average of the coupling constants being zero and leaving the rate unchanged. However, it is not clear to us what processes would cause the alignment of molecules in the experiments. There are fabrication techniques to construct such aligned molecular assemblies [60, 61] but they represent a departure from the microfluidistic cavity approach detailed in Ref. [12]. We hope that our work could motivate theoretical and experimental inquiries into this kind of disorder.

The effect of the positions of the molecules within the cavity has been investigated experimentally [45]. The experimental findings agree well with the current theoretical understanding presented here and in the main text. That is, local variations in the cavity field strength affect the couplings. We also know from Maxwell's equations that such vacuum electromagnetic fields may change sign as a function of position within the cavity. This is often neglected since observables such as the energy and the absorption/fluorescence spectrum are independent of the sign. Also, for a single molecule, the sign is always an irrelevant phase factor of the cavity field. The phase difference between different positions is important for the chemistry because one would expect that the molecules are distributed somewhat uniformly within the cavity. The average of the coupling constants should be zero for antisymmetric cavity modes and decreases as the order of the cavity mode is increased for symmetric modes. Thus, it would seem beneficial to limit the variation in the molecule positions by cavity design and/or using the lowest order cavity mode. The reader should be aware that this message is in contradiction with the published experiments in polaritonic chemistry: For instance, the second-order cavity mode was put in resonance with a vibrational mode in Ref. [17] and the tenth-order mode was used in Ref. [15]. It is an intriguing possibility that some effects would be, in fact, caused by a coupling to the highly detuned first cavity mode and, consequently, the interpretation of polaritonic chemistry as a resonant effect would be challenged.

In the presence of great position and orientation disorder, we expect no change in the rates. Nevertheless, if Rabi splitting is large enough and $\alpha > 1$, a phase separation based on the value of coupling constants is possible. If we imagine a case where we have somehow aligned the transition dipole moments and the relevant cavity mode is the second lowest order mode, the

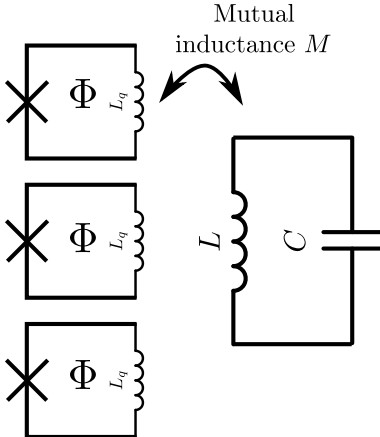

Figure 8: Electronic circuit model of an *LC* oscillator coupled to a set of flux qubits, where the Josephson junctions are marked with the triangle. There is a flux $\Phi = h/(2e) + \delta\Phi$ applied across each loop.

phase separation would manifest as a physical separation. The molecules on one side of the cavity would then go to another state than those on the other side. This could be achieved, alternatively, with two films of oriented dipole moments using the same measurement setup as in Ref. [45] so that the films are at the opposite sides of the cavity (that is, mirror-film-spacer-film-mirror).

## G.2 Coupled anharmonic *LC* oscillators

Although our work was initially motivated by polaritonic chemistry, our generic stochastic model is applicable to a large variety of systems. Here we illustrate how similar physics could be studied in an ensemble of superconducting flux qubits all coupled to the same cavity. We also show how in this setup one can in principle realize the alternating sign of the couplings.

Consider the electronic circuit drawn n Fig. 8. It describes an *LC* circuit coupled via mutual inductance with $N$ (ideally identical) flux qubits [62]. The figure depicts a simplified version of the flux qubit, but the idea can be generalized to the usually used configurations where the inductance $L_q$ is replaced by several Josephson junctions. Such systems have two macroscopic quantum states corresponding to clockwise and counterclockwise persistent currents, used as the logical qubit states. When the flux $\Phi \approx h/2e[1 + f/(2\pi)]$, the flux qubit Hamiltonian is

$$H = \frac{\hat{Q}_q^2}{2C_q} + \underbrace{E_J\left[b\phi^2 + \cos(\phi - f)\right]}_{V(\phi)}. \tag{G.1}$$

Here $\hat{Q}_q$ is the displacement charge operator of the capacitor with capacitance $C_q$. $\hat{Q}_q$ is canonically conjugate with the phase difference $\phi$ across the junction. They thus take the roles of canonical momentum and position in the electronic circuit. $E_J$ is the Josephson energy of the junction and $b = 2\pi^2 h^2/(2eLE_J)$ is a dimensionless parameter characterizing the inductor with inductance $L$. For $f \ll \pi$ and $b < 0.5$, the effective potential $V(\phi)$ has two minima corresponding to the two directions of the persistent current. The relative bias between the minima can be controlled with $f$ such that the two minima are degenerate for $f = 0$. For small $b$ and $f$, the minima are around $\phi_\pm \equiv \frac{\pm\pi}{1+2b} + f$, and both have eigenfrequencies $\omega_L = \omega_R \approx \omega_p\sqrt{(1+2b)}$, where $\omega_p = 2e\sqrt{E_J/C_q}/\hbar$ is the Josephson plasma frequency. Then, the effective barrier height is $E_b = \frac{1}{2}E_J(1+2b)[(\phi_+ - \phi_-)/2]^2 = \frac{\pi^2}{2}\frac{E_J}{1+2b}$.

Flux qubits operate in the regime where the potential barrier between the states is low, $E_b \sim \hbar\omega_{L/R}$, and thereby the tunnel coupling is large, providing coherent oscillations of the quantum state between the two minima. This limit is reached when the charging energy $E_C = e^2/(2C_q)$ is not much smaller than the Josephson energy $E_J$. On the contrary, we assume $E_J \gg E_C$ and thereby a high barrier, such that the transitions between the minima are rare and mostly driven by thermal noise in the environment of the system.

In such systems, the mutual inductance coupling of the individual qubits to a common $LC$ circuit has been used as a means to realize controllable coupling between the qubits, as they are tuned on and off resonance with each other [62]. However, there the resonance condition derives from the resonance of the qubit energy splitting, whereas we assume that the resonance is between the bistable energy minima. Let us then consider a mutual inductance coupling of the flux qubits to a common $LC$ circuit. We model this coupling in the limit where the qubits are in one of their minima, in which case we describe them as $LC$ oscillators. The Lagrangian of the system is thus

$$
\begin{aligned}
\mathcal{L} &= \frac{1}{2}L\dot{Q}_c^2 + \frac{1}{2}L_p\sum_{j=1}^{N}\dot{Q}_j^2 + \sum_{j=1}^{N}M_j\dot{Q}_j\dot{Q}_c - \frac{1}{2C}Q_c^2 - \frac{1}{2C_p}\sum_{j=1}^{N}Q_j^2 \\
&\equiv \frac{1}{2}\dot{\vec{Q}}^T\mathbf{L}\dot{\vec{Q}} - \frac{1}{2}\vec{Q}^T\mathbf{C}^{-1}\vec{Q},
\end{aligned}
\tag{G.2}
$$

where we have introduced capacitance $C_p$ and inductance $L_p$ such that $\omega_p = 1/\sqrt{L_p C_p}$ for the flux qubits. Note that this form of expression is possible because $\omega_L = \omega_R = \omega_p$; otherwise the state of the flux qubits affects the capacitance and inductance. The mutual inductance $M_j$ can be characterized by a dimensionless parameter $k_j \in [-1, 1]$ so that $M_j = k_j\sqrt{LL_p}$. Here, one can understand the possibility of a negative $k_j$ as a phase shift of the induced current; the sign depends on the details of the inductive coupling.

We define the conjugate momentum using the notation of the flux quantum $\Phi_0 = h/2e$ as

$$
\frac{\Phi_0}{2\pi}\varphi_i = \frac{\partial\mathcal{L}}{\partial\dot{Q}_i} = \begin{cases} L\dot{Q}_c + \sum_j M_j\dot{Q}_j, \\ L_p\dot{Q}_i + M_i\dot{Q}_c, \end{cases}
\tag{G.3}
$$

which can be also formally written as $\vec{\varphi} \propto \mathbf{L}\dot{\vec{Q}}$. With this definition, the left hand side represents magnetic flux. The Hamiltonian of the full systems is obtained by the Legendre transform. We find

$$
H_{\text{tot}} = \frac{1}{2}\left(\frac{\Phi_0}{2\pi}\right)^2\vec{\varphi}^T\mathbf{L}^{-1}\vec{\varphi} + \frac{1}{2}\vec{Q}^T\mathbf{C}^{-1}\vec{Q}.
\tag{G.4}
$$

The inversion of the inductance matrix $\mathbf{L}$ presents a difficult analytical problem. If the mutual inductances $M_j$ are large, that is $|k_j| \approx 1$, every element of the matrix $\mathbf{L}^{-1}$ is generally nonzero. However, we assume that they are small. To first order, the solution is obtained by noting that we can obtain the conjugate relation for $\varphi$ only in the zeroth order of $M$'s and substitute the result in the original Lagrangian $\mathcal{L}$ in Eq. (G.2) as it is first order in $M$'s. We find for an $LC$ circuit of resonant frequency $\omega_c = 1/\sqrt{LC}$

$$
\begin{aligned}
H_{\text{tot}} &\approx \frac{Q_c^2}{2C} + \sum_j\frac{Q_j^2}{2C_p} + \left(\frac{\Phi_0}{2\pi}\right)^2\left[\frac{1}{2}\frac{\varphi_c^2}{L} + \frac{1}{2}\sum_j\frac{\varphi_j^2}{L_p} + \sum_j\frac{k_j}{\sqrt{LL_p}}\varphi_c\varphi_j\right] \\
&\equiv \frac{P_c^2}{2} + \sum_j\frac{P_j^2}{2} + \frac{1}{2}\omega_c^2 X_c^2 + \frac{1}{2}\sum_j\omega_p^2 X_j + \omega_c\omega_p\sum_j k_j X_c X_j,
\end{aligned}
\tag{G.5}
$$

where, in the last step, we denoted $P_j = Q_j/\sqrt{C_p}$ and $P_c = Q_c/\sqrt{C}$ and correspondingly $X_c = \frac{\Phi_0}{2\pi}\sqrt{C}\varphi_c$ and $X_j = \frac{\Phi_0}{2\pi}\sqrt{C_p}\varphi_j$. Thus, in the lowest order approximation, the Hamiltonian corresponds to the potential we use in Appendix A with $\lambda_j^2 = k_j\omega_c\omega_p$. Note that both of these models are within the harmonic approximation, as we expanded the flux qubit Hamiltonian around a minimum point.

In other words, the cavity induced bifurcation could be observed in flux qubits by studying their collective evolution towards a state where even in the absence of a bias $f$, a majority of the qubits would be in the state with, say, clockwise circulating persistent current.

### G.3  Cold atoms

Recently, there has notable progress in the field of cold polar molecules [63]. Following the long tradition of optical trapping of cold molecules, this field provides an interesting opportunity to realize a controllable polaritonic system. This is because the many internal degrees of freedom in molecules allow for metastability in the energy, as described in this supplement and in the main text. Thus, in the future, it would seem possible to manufacture a system that is topologically the same as in polaritonics but with controllable coupling constants, and then investigate the noise activated processes or even quantum tunneling between the possible molecular states.

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
