# Peer review of "Cavity-induced bifurcation in classical rate theory"

_SciPost Physics, doi:SciPost Phys. 16, 025 (2024)_

## Round 3 · Referee Report · Anonymous · 2023-6-16

Report

I found this manuscript quite interesting, but not very well written, so I had to struggle abit in order to figure out what was done there.

Basically, the authors consider N classical two-level systems (TLSs), coupled to a single classical degree of freedom x. The first ingredient of the authors' model is the potential energy given by Eq.(4), where V_tot = V_tot(q_1,..., q_N, x), a function of N+1 coordinates. It has 2^N minima. In the absence of the coupling they are found trivially, but in the coupled system their positions shift, and the authors find these shifted positions and the corresponding energy of each minimum. Each minimum is labeled by a set of N discrete Ising-like variables which the authors denote by sign(q_i), instead of introducing a more compact notation s_i = +1 or -1. The energies of the minima are given by Eq.(6) in which V_tot = V_tot(s_1,...,s_N), a function of N discrete variables. More careful notations could save some effort for the reader here.

It took we a while to understand which quantities in Eq.(6) depend on {s_i} and which do not: while deltaN = sum_i s_i and Delta given by Eq.(5) obviously do, the index i at g_i and omega_i in Eq.(7) has a totally different meaning: while g_i depends only on i and not on s_i, the frequency omega_i is in fact omega(s_i), left or right. This is a rather misleading notation which obscures the meaning of Eqs. (27), (28) and the discussion around them which enables one to neglect the dependence of P on the configuration {s_i}.

In several places the authors mention transition dipole moments d'(+-a/2). Usually, the term "transition dipole moment" refers to an off-diagonal matrix element of the dipole moment between two stationary states. In the classical model used by the authors, the wave functions of the two states of each TLS have negligible overlap, so the off-diagonal dipole matrix element vanishes. Probably, the authors mean something else, but I did not understand what.

The second ingredient of the model are the rates of transitions between different minima. The only allowed transitions are those where just one TLS flips, so the system's configuration space can be conveniently represented as an N-dimensional hypercube. To each of its N*2^(N-1) edges the authors associate forward and backward transition rates based on the assumption of detailed balance. The authors try to express this via Eq.(3), but Eq.(3) as it is written is totally meaningless. Instead of V_tot(q_j<0) - V_tot(q_j>0) which totally unclear for V_tot(q_1,..., q_N, x) from Eq.(4), the detailed balance should contain V_tot(..., s_{j-1}, -1, s_{j+1},...) - V_tot(..., s_{j-1}, +1, s_{j+1},...), with V_tot defined in Eq.(6). Only after Eq.(6) one can discuss rates like in Eq.(3).

Strictly speaking, to describe the system's dynamics, a system of rate equations on a hypercube should be written for the probability to be in each of the 2^N potential minima, that is, the joint probability P(s_1,...,s_N). This is a linear system of 2^N equations for 2^N variables. However, the authors show that in the large N limit its stationary solution can be described by a mean-field self-consistent equation for Delta, derived in Appendix C. As far as I understand, to derive this self-consistent equation, one does not need rate equation at all; indeed, for any rates satisfying the detailed balance condition, the self-consistent equation can be found from the thermal Gibbs distribution with the energy V_tot(s_1,..,s_N) from Eq.(6). Thus, the phase transition arising in this model is of EQUILIBRIUM nature.

The rate equations are only needed to describe the system's relaxation towards equilibrium. In the large N limit, the full 2^N by 2^N linear system can be reduced to Eq.(40) when all couplings g_i are equal. When all g_i are different, the task is much more complicated, and the authors perform a brute force stochastic simulation in Appendix E. I wonder, is it possible to proceed analogosly to Appendix C and derive a closed kinetic equation for the distribution function p(g), the probability for a TLS whose coupling g_i is in the interval g < g_i < g + dg to have s_i = -1? Such a kinetic equation might be tractable.

In the authors' model the cavity mode is treated classically. Strictly speaking, this is valid only when the reorganization energy is much larger than the mode frequency (that is, the shifted ground state contains many photons). I did not see this condition checked anywhere in the paper. At the same time, in Sec. 5 the authors obtain a similar Ising energy functional V_tot from the quantum Dicke-like model. However, (i) this only works in the absence of the energy difference between the two states of each TLS, and (ii) transition rates will be dressed by Debye-Waller factors (overlaps between the shifted cavity ground states). So I still doubt that the presented calculation really applies to cavity polaritons.

Above I summarized my understanding of the paper (let the authors correct me if I missed something). Based on this understanding, I get the following impression about the paper's results.

1. I do not see any polaritonic chemistry here. I only see the good old ferroelectric transition, the same as found in the equilibrium Dicke model. The case of TLSs not being identical has been treated in the literature [Phys. Rev. B 64, 235101 (2001); Nucl. Phys. B 919, 218 (2017)]. That case was not exactly the same as here, since here the authors include randomness in the dipole moments instead of randomness in TLS energies, but I do not see any qualitative difference. So the authors' claim in the introduction that they predict a phase transition seems to be an overstatement: in fact, they study the same phase transition that was predicted long ago, but they do it in a different model.

2. I have strong doubts about the applicability of the authors' model to polaritonic systems. Besides the classical treatment of the cavity mode that I already mentioned, I see one more problem. If the TLSs couple to the cavity mode's electric field via their dipoles, they should also couple to each other directly via the Coulomb dipole-dipole interaction. Why is it not included?

3. The circuit QED implementation discussed in Appendix G.2. resembles very much the one proposed in Phys. Rev. Lett. 104, 023601 (2010). Also, superconducting circuits typically have low dissipation, so quantum Hamiltonian treatment should be more relevant for such systems than the classical dissipative model studied in the present paper. So we are back to the model applicability problem.

4. The only part of the paper that looks to me somewhat novel with respect to the earlier literature is the study of the relaxation dynamics presented in Appendices E, F. But its results could be expected without any calculations. Yes, the system relaxes towards equilibrium. The relaxation time scale naturally matches the rates that were plugged into the model. So I do not see what one can learn from these results.

To conclude, I see too many issues with the current version of the manuscript to recommend it for publication.

  • validity: -
  • significance: -
  • originality: -
  • clarity: -
  • formatting: -
  • grammar: -

Author:  Kalle Kansanen  on 2023-09-19  [id 3990]

(in reply to Report 1 on 2023-06-16)

We thank for the numerous good suggestions regarding the presentation of the manuscript. The overview touches upon all the important parts of the manuscript except the description of how the initial relaxation or reaction rate is modified. While there are some points to be clarified, we agree upon this overview until the discussion around the classicality of the cavity mode. That part of the report mostly concerns the way we have presented the theory, not so much the actual results. On the other hand, the interpretations of the manuscript that follow we do not share.

Our initial goal was to understand what kind of polaritonic chemistry there can be when the typical chemistry tools such as transition-state theory suggest that no effects should be observed in large ensembles of weakly coupled molecules. We admit that we have probably misused the term "polaritonic" in this regard, as the word polariton refers to the hybrid mode between the cavity field and the molecules, and our result, while starting from the same setting as the usual polaritonic chemistry papers, does not rely on the avoided crossing between the cavity and molecular resonances. The main idea was that perhaps one has to be careful in constructing the rate equations themselves, which is why our manuscript is framed in that language. This seemed to us like a general physics question of wider interest. We found interesting, for instance, that there can be a phase transition which does not affect the reaction rates. After establishing the overall picture in the rate equations, we noticed a similarity to the phase transition in the Dicke model discussed in Sec. 5. However, unlike many of the reviewers seem to assume, these phase transitions are not the same. In particular, the phase transition in the reaction rate theory has an additional parameter compared to the Dicke model, namely the (effective) height of the reaction barrier.

We have now modified the text to clarify the precise connections to the polaritonics and to the superradiant transition in the Dicke model.

Next, we comment on specific parts of the report.

Each minimum is labeled by a set of N discrete Ising-like variables which the authors denote by sign(q_i), instead of introducing a more compact notation s_i = +1 or -1. The energies of the minima are given by Eq.(6) in which V_tot = V_tot(s_1,...,s_N), a function of N discrete variables. More careful notations could save some effort for the reader here.

We agree on the suggestion, and the sentiment of the following paragraph that some notations were left too implicit. The functional dependencies are now stated more clearly throughout the manuscript, especially sections 2 and 3 and appendices A, B and F.

Usually, the term "transition dipole moment" refers to an off-diagonal matrix element of the dipole moment between two stationary states.

This is to our understanding standard nomenclature in the dipole gauge, see for instance Eqs. (1) and (4) in this JCPL by Yang and Cao. Here, the relevant states are the vibrational states in the right and left well which are approximately like harmonic oscillators. Another way to think of the terminology is that if the dipole function $d_j(q_j)$ of Eq. (4) is a constant function, there is no light-matter coupling at all in the model.

[...] the self-consistent equation can be found from the thermal Gibbs distribution with the energy V_tot(s_1,..,s_N) from Eq.(6).

This is true, and an important point we left implicit. To clarify this, we have added a sentence "We note that our approach is consistent with thermodynamics by construction, and hence the stationary properties could also be found by treating the light-matter system as a canonical ensemble and solving its thermal distribution" before Sec. 4. However, as we not only want to discuss the stationary states but also the transients, we prefer to resort to the rate equation model.

I wonder, is it possible to proceed analogosly to Appendix C and derive a closed kinetic equation for the distribution function p(g), the probability for a TLS whose coupling g_i is in the interval g < g_i < g + dg to have s_i = -1? Such a kinetic equation might be tractable.

This is definitely a good idea. One can indeed derive a kinetic equation with the same approach. However, the problem is that one would need the time evolution of the order parameter $\Delta$, and in turn, the kinetic equation for $\Delta$ depends on the integral over the distribution function. The coupled equations do not close in any straightforward way. It is mentioned in the paragraph containing Eqs. (11) and (12).

In the authors' model the cavity mode is treated classically. Strictly speaking, this is valid only when the reorganization energy is much larger than the mode frequency (that is, the shifted ground state contains many photons). I did not see this condition checked anywhere in the paper. At the same time, in Sec. 5 the authors obtain a similar Ising energy functional V_tot from the quantum Dicke-like model. However, (i) this only works in the absence of the energy difference between the two states of each TLS, and (ii) transition rates will be dressed by Debye-Waller factors (overlaps between the shifted cavity ground states). So I still doubt that the presented calculation really applies to cavity polaritons.

We fail to see how exactly it is important that the cavity mode would be treated classically. Since the cavity's contribution to the energy is linear and we find the effective interaction it mediates, it would seem that we would end up to the same result with a quantum mechanical cavity mode. We do not understand what is meant here by ''reorganization energy'' (which is a concept used at least in charge transfer and open quantum systems) and hence the argued validity condition.

To clarify: the most important effect of the cavity is to mediate the interaction between the dipoles. To some extent, this can be viewed as the cavity field providing an effective quantum vacuum to the molecules. It is hence fully compliant with quantum mechanics. Whether the effect is quantum mechanical or classical is a semantic problem. One can compare this to the effective interaction between electrons mediated by phonons, forming the basis of the theory of conventional superconductivity.

We do not fully understand the second part of this comment, related with the Ising energy functional, energy differences between the states, and the Debye-Waller factors. First, the Ising functional is presented only as a reference point, as it does not represent the same physics as we found in the transition state theory. We also fail to see the relevance of Debye-Waller factors in our model, and do not know which shifts of the cavity ground states the referee talks about. In other words, it is unclear if the referee here is questioning the starting model (e.g., Eq. (4)), or the approach used to describe the system behavior. We note that we assume an ensemble of same molecules (i.e., the same microscopic parameters), but allow for randomness in their positions and orientations, along with the stochasticity of the individual transition states of each molecule. Is this setting not well-defined enough?

1. I do not see any polaritonic chemistry here. I only see the good old ferroelectric transition, the same as found in the equilibrium Dicke model. The case of TLSs not being identical has been treated in the literature [Phys. Rev. B 64, 235101 (2001); Nucl. Phys. B 919, 218 (2017)]. That case was not exactly the same as here, since here the authors include randomness in the dipole moments instead of randomness in TLS energies, but I do not see any qualitative difference.

Physically, the dipole moments play a fully different role than the eigenfrequency of the molecular or atomic mode. In the picture of cavity-mediated dipole-dipole interaction, the dipole moments determine if the effective interaction is attractive or repulsive in nature. Mere changes in eigenfrequency will not achieve this. We also point out that although seemingly analogous, the Dicke model transition is different than the transition in the rate equation model, because the latter has an extra parameter: the height $E_b$ of the potential barrier in the potential energy surface. As we point out in Sec. 5, the transition temperature in the rate equation model is $2 E_b/\omega_b$ times larger than in the (simple) Dicke model.

We admit that the connection between the two models (and the fact that they are not the same) was quite unclear in the manuscript, so we have clarified it in the beginning of Sec. 5.

2. I have strong doubts about the applicability of the authors' model to polaritonic systems. Besides the classical treatment of the cavity mode that I already mentioned, I see one more problem. If the TLSs couple to the cavity mode's electric field via their dipoles, they should also couple to each other directly via the Coulomb dipole-dipole interaction. Why is it not included?

As we already mention above, we disagree with the claim of disregarding some quantum effects related with the presence of the cavity.

The direct coupling via Coulomb dipole-dipole interaction is definitely possible. However, we assume that such coupling will be too weak to induce the bifurcation transition, whereas the cavity mediated coupling can be much stronger. Taking into account everything amounts to calculating the Casimir-Polder-van der Waals interaction between the molecules inside a cavity. The detailed form of this interaction will then depend on the precise locations of the molecules inside the cavity and with respect to each other, and their dipole orientation not only with respect to each other, but also to the cavity electric field, and on the type of the electromagnetic cavity. After specifying the system completely this calculation is in principle possible, but it is quite tedious and quite definitely outside the scope of the present work. We fear that its technical details would also obfuscate the big picture related with the bifurcation transition. Our model should therefore be considered as the relevant toy model indicating the possibility of the bifurcation dynamics, and the relevance of the orientation of the molecular dipoles.

We have now clarified these issues below Eq. (4).

3. The circuit QED implementation discussed in Appendix G.2. resembles very much the one proposed in Phys. Rev. Lett. 104, 023601 (2010). Also, superconducting circuits typically have low dissipation, so quantum Hamiltonian treatment should be more relevant for such systems than the classical dissipative model studied in the present paper. So we are back to the model applicability problem.

In fact, our motivation of writing Appendix G.2 stemmed from the need of understanding the physics of the sign of the dipole moment. In the case discussed in Appendix G.2 this sign is directly connected with the relative windings of the inductors mediating the coupling.

The model in the quoted reference indeed looks at first quite similar to the one we have in Appendix G.2, and is an interesting manifestation of a model quite analogous to the Dicke model. However, we are in a different regime of parameters: in that PRL, similar to the usual flux qubit variants (as in Ref. [63]), the relevant parameter of interest is the coupling energy between the two wells of the "flux potential" (this parameter is $\omega_F$ in that PRL). The resonance condition directly compares to this energy. In our case, we assume this energy to be quite small (in other words, the potential barrier between the wells is high), and the cavity frequency is compared only to the resonance frequencies of the two harmonic potentials inside the wells.

We disagree on the necessity of using a fully quantum mechanical treatment for any superconducting circuit. The (classical) transition rate theory is valid in the case where after each transition the system thermalizes to the local equilibrium state before a new transition takes place. In other words, when the transition rate is much smaller than the internal dissipation rate, the classical description is accurate. In the circuit model, the transition rates are effectively controlled by the Josephson energy, so we see no fundamental reason why one could not reach such a parameter regime. As is painfully known in the superconducting qubit community, even superconducting circuits exhibit finite dissipation (and thereby dephasing/relaxation).

4. The only part of the paper that looks to me somewhat novel with respect to the earlier literature is the study of the relaxation dynamics presented in Appendices E, F. But its results could be expected without any calculations. Yes, the system relaxes towards equilibrium. The relaxation time scale naturally matches the rates that were plugged into the model. So I do not see what one can learn from these results.

For us, the initial rate modification, Eq. (14), is not obvious but a result of some effort, so we would be delighted to know how it can be derived without any calculations. In polaritonics, at least, we often encounter a view where only the Rabi splitting should matter, as it determines the energy spectrum, so having the rate modification to be proportional to the average of the light-matter coupling constants is not something we understood before starting.

---

## Round 3 · Referee Report · Anonymous · 2023-6-30

Report

The manuscript presents a theoretical analysis of the Dicke model, an ensemble of identical two-state systems coupled to one electromagnetic mode. Although the authors formulate the model in a way that is technically more general than the Dicke model (two-state systems are replaced with the bistable systems), the main results of the manuscript, as emphasized in the abstract, symmetry-breaking phase transition, is obtained in the regime where the model used in this work is identical with the Dicke model. In this regime, the element of novelty of this manuscript is that the dynamics of the two-state systems is described with the rate equation. Since the rate equation implies irreversible, i.e., ``incoherent’’ dynamics of the two-state systems, such a description can in principle be very different from the more standard coherent regime, and the manuscript can be suitable for publication. One major issue, however, needs to be clarified before publication. The phase transition discussed by the authors in the incoherent regime is the same phase transition that was discussed over the years for the Dicke model. By now, it seems more or less well established that the phase transition does not exists in the context of that model (as discussed, e.g., in Ref. 54 of the manuscript). Although the authors attempt to explain away this contradiction with their results, the arguments used (see Sec. 5 of the manuscript) clearly do not make sense, because the ``strong assumptions’’ underlying the proofs of the absence of the phase transition are the same assumptions used by the authors in this work. The authors should provide a clear and convincing resolution of this issue. There seems to be only two options in this regard.
1. Either the incoherent nature of the two-state dynamics makes the situation so different that the previous arguments for the absence of the phase transition do not apply. In this case, the authors should stress explicitly that their description applies only to incoherent systems, and give an estimate of how strong the decoherence should be for their description to apply. The large temperature mentioned in the manuscript is not sufficient to make the rate equations valid.
2. The second option is that the authors’ description based on the potential energy (4) is flawed in this respect, and can not be applied to the system the authors have in mind, electric dipoles interacting with a mode of electromagnetic cavity. This option seems to be more probable for the following reason. The main physical effect driving the phase transition discussed in this work is the effective dipole-dipole interaction induced by the cavity that reduces the energy of the states in which all the dipoles are aligned. This interaction is of the second order in the dipole-cavity interaction in Eq. (4), the fact that requires one to also include into Eq. (4) the terms of the second order in the dipole-cavity interaction. In the case of an electromagnetic mode, there are always quadratic terms in the standard ``textbook’’ Hamiltonian that describes interaction of charged particles with an electromagnetic field. In this respect, the reference to paper [37] used by the authors to neglect such terms looks unjustified, since the coupling to a cavity mode has no connection to the direct Coulomb dipole-dipole interaction, and because of this, effects of the coupling to a cavity mode cannot be cancelled out by this interaction. In any case, since the authors claim the appearance of the phase transition as their main result, and since the existing literature (e.g., Refs. 37 and 54) has directly contradicting conclusions concerning this phase transition, the authors should make a serious effort to formulate a convincing and logical position on this issue.
There is also a minor problem with the text of the manuscript that should be corrected before possible publication. The term ``collective phase transition’’ used in the manuscript is very confusing, because it implies that there are also phase transitions that are not ``collective’’. There are none. All phase transitions are collective by their very nature, and the term should be changed to simply ``phase transition’’.

  • validity: -
  • significance: -
  • originality: -
  • clarity: -
  • formatting: -
  • grammar: -

Author:  Kalle Kansanen  on 2023-09-19  [id 3991]

(in reply to Report 2 on 2023-06-30)

We appreciate the thorough reading of our manuscript.

The question of equilibrium phase transitions in QED is something we stumbled upon, not knowing its fifty year history beforehand. And we agree, it is an important and interesting question. Moreover, as discussed in our response to referee 1, the superradiant transition in the Dicke model is not the same transition as what we found within the transition state theory.

While the statement

By now, it seems more or less well established that the phase transition does not exists in the context of that model (as discussed, e.g., in Ref. 54 of the manuscript). Although the authors attempt to explain away this contradiction with their results, the arguments used (see Sec. 5 of the manuscript) clearly do not make sense, because the ''strong assumptions'' underlying the proofs of the absence of the phase transition are the same assumptions used by the authors in this work.

makes it sound as if the issue is fully resolved, the more we looked into it, the more nuanced and difficult it appears. Indeed, Ref. 54 claims that there can be no phase transitions, yet Ref. 55 from the same authors published only a year after Ref. 54 says that there is a phase transition. (Note that these reference numbers point to the manuscript before resubmission, v3 in arXiv.) The relaxed assumption in that case was that the EM field varies over the molecules. This means that, in Ref. 54, the molecules are effectively in the same location within a cavity - clearly not a valid assumption in most experiments. Therefore, our reading of the literature so far is that QED predicts equilibrium phase transitions.

An added difficulty is that the dipole gauge is explicitly derived from the Coulomb gauge assuming this "long wavelength" -limit where the position of molecules is fully neglected. This means that the EM vector field is also position independent. When one relaxes this assumption, the EM vector field depends on position, and moreover, does not commute with itself at different positions. This makes the unitary transformation to the dipole gauge very complicated, so much so that we have not seen it in the literature. As a consequence, comparing results in different gauges is not straightforward.

There is also a minor problem with the text of the manuscript that should be corrected before possible publication. The term ''collective phase transition'' used in the manuscript is very confusing, because it implies that there are also phase transitions that are not ''collective''. There are none. All phase transitions are collective by their very nature, and the term should be changed to simply ''phase transition''.

The use of the word collective is indeed tautological. Our main hope was to differentiate our work from those, e.g., in the ultrastrong coupling regime. Judging from the comment, it was not a successful choice and has been removed everywhere. We renamed Sec. 4 as "Cavity-induced phase transition and collective dynamics" to better reflect the content of Sec. 4.3.

---

## Round 3 · Referee Report · Anonymous · 2023-7-1

Report

The manuscript "Cavity-induced bifurcation in classical rate theory" present a classical treatment of a collection of model molecules coupled to a cavity mode. While the other referees have already raised some relevant concerns, I believe that there are several additional issues that affect the generality and applicability of the model and results. These are:
-) Only a single cavity mode is considered. This can capture a significant part of the physics of excited states resonantly coupling to that mode, and thus is widely used to describe polariton properties and dynamics. However, the effects discussed in the manuscript do not depend on resonance behavior as far as I understood, and all modes are expected to contribute. Restricting the model to just a single cavity mode is thus a severe approximation. This is the case especially for the planar cavities used in the experiments on vibrational strong coupling that the manuscript is focused on. Such cavities support modes at any in-plane momentum and at any mode order for the discrete out-of-plane direction. Including all modes is the same as solving the electromagnetic problem (Maxwell's equations) directly, and since dynamics do not play a role here, I believe this would just lead to electrostatics (i.e., the mirrors would produce image dipoles).
-) The molecular dipoles are assumed to be aligned to the electric field of the cavity. In typical experiments, they are disordered and point in arbitrary directions. While for a single-mode description, non-perfect orientation can be included by adjusting the coupling strengths, when all modes are taken into account, the orientation of the dipoles becomes much more important.

Given the above points, I believe that the model and results presented in the manuscript are not suitable for SciPost Physics. I do believe the presented model is interesting and can capture some aspects of the physics of the system, but I think it is too limited to be of general interest. I would be happy to support publication in a more specialized journal.

  • validity: -
  • significance: -
  • originality: -
  • clarity: -
  • formatting: -
  • grammar: -

Author:  Kalle Kansanen  on 2023-09-19  [id 3992]

(in reply to Report 3 on 2023-07-01)

In our view, the broader physics interest from our work is the rate theory construction of interacting systems (mediated by a cavity) and how collective effects arise in that framework. We hope that our response to the limitations of this approach could affect your consideration.

Only a single cavity mode is considered. This can capture a significant part of the physics of excited states resonantly coupling to that mode, and thus is widely used to describe polariton properties and dynamics. However, the effects discussed in the manuscript do not depend on resonance behavior as far as I understood, and all modes are expected to contribute. Restricting the model to just a single cavity mode is thus a severe approximation.

In principle, taking into account many modes phenomenologically does not really change the picture. This is probably best exemplified by the section 5 where we explicitly take into account infinitely many modes of the cavity. While we do not discuss this, it is clear from Eq. (16) that we change the single-mode result to a sum over all the modes. The same would happen in the rate theory approach. But, based on the second point,

The molecular dipoles are assumed to be aligned to the electric field of the cavity. In typical experiments, they are disordered and point in arbitrary directions. While for a single-mode description, non-perfect orientation can be included by adjusting the coupling strengths, when all modes are taken into account, the orientation of the dipoles becomes much more important.

it seems to us that one would hope for a full calculation starting from Maxwell's equations to determine all the constants that are currently merely phenomenological in terms of physical parameters of the cavity. Doing this and cutting the infinite sums is something we have thought of, but it is a difficult task we could not finish for this work. While this result would be fantastic, our intuition is that it doesn't change the rate theory approach or the qualitative results (e.g., disorder in the dipole moments removes the cavity effect on the initial rates). We note that the issue of self-interaction terms in the QED description we have discussed with another referee is relevant for such a calculation.

In response to this comment, we have added the text "Moreover, the model containing only a single cavity mode is chosen for simplicity. Generalizing the results to the case of several cavity modes is relatively straightforward and does not lead to qualitative changes in the results." in the paragraph below Eq. (4).

---

## Round 4 · Referee Report · Anonymous (Referee 1) · 2023-10-3

Report

I thank the authors for their clarifications, but unfortunately I am still not satisfied. I guess, my main problem is that Eqs.(3) and (4) are not sufficient for me to define the model and to decide on its applicability to various physical systems. I also need the kinetic energy and the associated hierarchy of energy scales. Since the kinetic energy is not mentioned at all in the model, my first guess was that all degrees of freedom are in the overdamped limit, the frequency is a small energy scale, the masses are heavy, and the wave function is well localized in the coordinates x,q_i at all times; this is what I meant by the classical treatment. Apparently, my guess was wrong, and the authors have in mind something else. If we adopt the fully quantum picture for all degrees of freedom, like in the Dicke-like model in Sec. 5, then transitions flipping sigma^i_z must be due to some perturbation proportional to sigma^i_x or sigma^i_y. In this case, since the unitary transformation shifts the a_alpha modes to different ground states for different sigma^i_z, the transition rate must involve the overlap between these shifted ground states (the Debye-Waller factor), which the authors do not consider. So my current guess about the authors' model is that the cavity mode has high frequency, so it remains in the instantaneous ground state which adiabatically follows the slow overdamped degrees of freedom q_i (Born-Oppenheimer approximation). In this case, the frequencies must be strongly different and one cannot speak about polaritons.

I still think that the transition found by the authors is physically indistinguishable from the Dicke phase transition. The authors seem to have agreed that it has nothing to do with the kinetics of the system, but is an equilibrium property. Essentially, it is an instability of the system "dipoles + cavity mode". One can use different models for dipoles and their coupling, they may be richer than the origianl Dicke model and have more parameters, but essentially the phenomenon is the same, if one tries to define the transition in model-independent terms.

Originally, I thought the issue of "transition dipole moment" to be a purely terminological one, but now I start to suspect that it is a manifestation of some deeper confusion. The term "transition matrix element" is a standard and general one in quantum mechanics, and it denotes an off-diagonal matrix element of some operator between two stationary states of the system, when this operator acts as a perturbation. It has nothing to do with the dipole gauge. In the authors' construction, d(q) is the diagonal matrix element, as far as I can understand. For this reason, to me the authors' construction seems different from Van-der-Waals/Casimir-Polder interaction because the latter is determined by the off-diagonal dipole matrix elements, rather than by the static (diagonal) dipoles. For the same reason, I do not see any relation between the dipoles introduced by the authors and the Rabi frequency, because the latter is determined by the off-diagonal dipole matrix elements. Also, if the frequency of the mode is strongly different from that of the q_i variables, one cannot speak about splitting.

I see no reason why the Coulomb dipole-dipole interaction would be weaker than the cavity-induced one. Usually it is the opposite, unless the cavity mode is at resonance with a molecular transition. But the effect considered here is non-resonant. If the main reason to neglect Coulomb is not to obfuscate the big picture, then let the authors study the big picture, but not claim its relevance to molecules in a cavity.

Concerning the potential realization in superconducting circuits, I agree with the general statement that the authors' construction is valid when the transition rate is much smaller than the internal dissipation rate, but my point is that in superconducting circuits it is usually the opposite, especially in qubits. So to claim applicability to superconducting circuits, the authors must check if realistic circuits with suitable hierarchy of scales really exist.

I admit that Eq.(14) is a valid non-trivial result. But I do not see much of its manifestation in the relaxation curves shown in Figs. 3,4, and 6. All these curves relax on the time scale ~ 1/Gamma_0.

To conclude, the paper still apears to be very vague to me, applicability of the presented results to real physical systems looks doubtful, so I cannot recommend it for publication.
  • validity: -
  • significance: -
  • originality: -
  • clarity: -
  • formatting: -
  • grammar: -

Author:  Kalle Kansanen  on 2023-12-11  [id 4181]

(in reply to Report 1 on 2023-10-03)

We thank the referee for their critical comments which have helped us to clarify the presentation of the manuscript. We recognize the issue of setting up the discussion in a concise and understandable way, choosing the strategy of explaining a much simpler textbook case in Sec. 2 and expanding from that. We have further clarified the manuscript on this starting point which we had taken too much for granted. The physics of the manuscript remains the same, with the model limitations we have discussed also with the other referees. We then respond to the comments in detail.

I thank the authors for their clarifications, but unfortunately I am still not satisfied. I guess, my main problem is that Eqs.(3) and (4) are not sufficient for me to define the model and to decide on its applicability to various physical systems. I also need the kinetic energy and the associated hierarchy of energy scales. Since the kinetic energy is not mentioned at all in the model, my first guess was that all degrees of freedom are in the overdamped limit, the frequency is a small energy scale, the masses are heavy, and the wave function is well localized in the coordinates x,q_i at all times; this is what I meant by the classical treatment. Apparently, my guess was wrong, and the authors have in mind something else. [...]

Our results are valid when thermal activation over the potential barrier dominates over the quantum tunneling rate, which is the proper limit either in the limit of a large mass or a high internal dissipation rate (large linewidth) - as the referee also correctly guessed. One of us has also studied the direct polaritonic effect on the quantum tunneling rate (Phys. Rev. B 107 (3), 035405 (2023)). The conclusion there is that the polaritonic effects on the overall tunneling rate vanish in the large-$N$ limit.

The referee's confusion seems to be related with our claim that the cavity effect can still be a quantum effect. Namely, the interaction between the dipoles mediated by the cavity can be obtained also from a quantum treatment.

In other words, we assume that the transition rate (times $\hbar$) is a negligible energy compared to the intrawell energy scales. We have clarified this further below Eq. (2) in the manuscript. On the other hand, the cavity induced coupling between the molecules is non-negligible compared to the intrawell energy scales in such a way that the parameter alpha defined below Eq. (10) can be larger than unity. In this case the presence of the cavity does not break the assumptions behind the thermal activation model.

I still think that the transition found by the authors is physically indistinguishable from the Dicke phase transition. The authors seem to have agreed that it has nothing to do with the kinetics of the system, but is an equilibrium property. Essentially, it is an instability of the system "dipoles + cavity mode". One can use different models for dipoles and their coupling, they may be richer than the origianl Dicke model and have more parameters, but essentially the phenomenon is the same, if one tries to define the transition in model-independent terms.

We believe that the refereen's confusion between our stochastic model and the phase transition in the Dicke model originates from the same misunderstanding of the hierarchy of energy scales. Nevertheless, there is certainly a connection to the Dicke model and its phase transition after some reinterpretation of our starting point. This is our reasoning for writing Sec. 5.

Originally, I thought the issue of "transition dipole moment" to be a purely terminological one, but now I start to suspect that it is a manifestation of some deeper confusion. [...]

In the molecular systems which we have in mind, the function $d(q)$ is indeed a diagonal matrix element in terms of the dipole operator and the electronic ground state which in turn depends parametrically on the classical $q$ (e.g., distance of two nuclei). As mentioned in Report 3, it is the variation of $d(q)$ with $q$ that gives the vibrational transition moment as nondiagonal matrix elements of the vibrational states. That is, in fully quantum mechanical description, $q$ is promoted to an operator. One certainly can derive this in any gauge they wish but it is the most straightforward to grasp in the dipole gauge.

I see no reason why the Coulomb dipole-dipole interaction would be weaker than the cavity-induced one. Usually it is the opposite, unless the cavity mode is at resonance with a molecular transition. But the effect considered here is non-resonant. If the main reason to neglect Coulomb is not to obfuscate the big picture, then let the authors study the big picture, but not claim its relevance to molecules in a cavity.

We see the situation as follows. In our simple model, we provide a prediction for the bifurcation parameter $\alpha$ below Eq. (10) using Eq. (7) for the parameter $P$. Had we done the complete microscopic calculation that includes the Coulomb dipole-dipole interaction, the relation between $\alpha$ and the microscopic parameters would change in a way that reflects the assumptions done in this model (say, the molecule positions within the cavity matter, and also their dipole moment directions not only with respect to each other, but also with respect to the main cavity axis). Nevertheless, the presence of the bifurcation transition would not change. The main point in this work is to illustrate this bifurcation transition and not the details of the dipole-dipole interaction.

[...] the transition rate is much smaller than the internal dissipation rate, but my point is that in superconducting circuits it is usually the opposite, especially in qubits. So to claim applicability to superconducting circuits, the authors must check if realistic circuits with suitable hierarchy of scales really exist.

The intrawell relaxation (i.e., the internal quality factor) in superconducting circuits can be controlled by the presence of extra shunts. This is the standard way to remove hysteresis from Josephson junction circuits. In effective circuit models such as the RCSJ model (see "The Physics of Nanoelectronics" by T.T. Heikkilä, Oxford University Press 2013, Ch. 9), this thus means adding an extra resistor element to the circuit. Hence getting to "our" limit of low quality factor is straightforward; much easier than getting to the "usual" flux qubit limit mentioned by the referee.

To give one concrete example, Gang Li et al. in Chin. Phys. B 27, 068501 (2018) describe measurements on a much more complicated 'flux qubit' (operated out of the typical flux qubit regime) than discussed here. Their Eq. (1) is equivalent with our Eq. (46) (note that our definitions of $\phi$ differ by a factor of $2\pi$) but they may tune the constants with external fields. They report reaching $b \approx 0.3$ and $f = 0$, putting our works in the same regime. Furthermore, they explicitly measure the thermal activation rates and detailed balance relation.

I admit that Eq.(14) is a valid non-trivial result. But I do not see much of its manifestation in the relaxation curves shown in Figs. 3,4, and 6. All these curves relax on the time scale ~ 1/Gamma_0.

Does the referee claim that the curves corresponding to different distributions of the dipolar coupling $g$ in Fig. 3 are all the same? To us they look different because they are not all overlapping. This difference comes from the different variances of $g$, thereby illustrating it exactly.

It is true that the figures of our manuscript do not effectively visualize the change in the thermalization time $t_{th}$, defined for instance from $[N_L(t_{th}) - N_L(0)]/[N_L(\infty) - N_L(0)] = e^{-1}$. Instead, we focus on the initial rates and the steady state values, both of which are obviously experimentally relevant. As an approximation, $t_{th} \approx 1/\Gamma_{LR}$, showing that since the rate modifications are of the order of unity so are the modifications of the thermalization times. As far as our reading of the literature goes, all the claimed effects in polaritonic chemistry are also of the order of unity.

---

## Round 4 · Referee Report · Anonymous (Referee 2) · 2023-10-9

Report

The authors provided a satisfactory response to my main technical question regarding the omission of the dipole self-energy terms in the starting equation (4) for potential energy.
In my view, the manuscript in its present form is suitable for publication. The fact that the ``Dicke model’’ has a phase transition in the regime when the dynamics of the two-state systems in the model is classical is new, and at least one of the suggested experimental realizations of this proposal based on the bistable dynamics of flux ``qubits’’ also looks technically sound and interesting.

---

## Round 4 · Referee Report · Anonymous (Referee 3) · 2023-11-13

Report

The authors have responded to all the points raised by the referees. While I agree that some of the issues that one of the other referees pointed out could be clarified, I believe that the paper is still more complete than many other works in the field and that it is worth publishing and appropriate for SciPost Physics. I therefore recommend publication.

Requested changes

1- The issue of the (transition) dipole moments mentioned by one of the other referees seems to be a confusion between the adiabatic "electronic" molecular dipole d(q), which is indeed the q-dependent diagonal expectation value, and the vibrational transition dipole between the (quantised) vibrational states within the adiabatic potential (which to lowest order is proportional to d'(q0), i.e., the spatial derivative of the adiabatic dipole, since d(q) = d(q0) + d'(q0)*(q-q0)). This is used implicitly in the paper, but not well-explained and should be clarified.

  • validity: -
  • significance: -
  • originality: -
  • clarity: -
  • formatting: -
  • grammar: -

Author:  Kalle Kansanen  on 2023-12-11  [id 4182]

(in reply to Report 3 on 2023-11-13)

We thank for the further clarification on the dipole moment function. We have also added the mentioned linear approximation to the main text, as it was previously only given in the Appendix A in a parametrized form [Eq. (20)].

---

## Round 4 · List of Changes

• Throughout the manuscript, we have updated the mathematical notation so that the states are described with variables s_i obtaining values +1 or -1, and it is easier to see the quantities that depend on these states.
  • Throughout the manuscript, we have removed the tautological word 'collective' in connection with 'phase transition'. Similarly, the word 'polaritonic' is removed in several places in order to avoid confusion, and in general the connections between our work and polaritonics should now be more clear.
  • The light-matter coupling model and its assumptions are further discussed below Eq. (4), including the exclusion of direct dipole-dipole interaction.
  • The meaning of the transition dipole moment in the context of our manuscript is elaborated in Sec. 3.
  • Equation (7) expresses the approximation used in the main text.
  • The consistency of our approach to thermodynamics is elaborated at the end of Sec. 3.
  • The beginning of Sec. 5 presents more clearly the connection between the rate theory approach and the Dicke model.
  • Two references added, Refs. [20] and [43], shifting reference numbers.

---

## Round 5 · List of Changes

• Extended paragraph below Eq. (2) to discuss the physical applicability of classical rate or master equations
  • Added a further explanation on the dipole moment function below Eq. (4) and its first-order expansion around the potential minima in the following paragraph
  • Added Ref. 25 as an example of cavity effects on quantum tunneling

---

## Editorial Decision

published